



# Machine learning methods for stream water temperature prediction

Moritz Feigl[1,*], Katharina Lebiedzinski[1,*], Mathew Herrnegger[1], and Karsten Schulz[1]

[1]Institute for Hydrology and Water Management, University of Natural Resources and Life Sciences, Vienna, Austria
[*]These authors contributed equally to this work.

**Correspondence:** Moritz Feigl (moritz.feigl@boku.ac.at)

**Abstract.** Water temperature in rivers is a crucial environmental factor with the ability to alter hydro-ecological as well as socio-economic conditions within a catchment. The development of modelling concepts for predicting river water temperature is and will be essential for an effective integrated water management and the development of adaptation strategies to future global changes (e.g. climate change). This study tests the performance of 6 different machine learning models: step-wise linear regression, Random forest, eXtreme Gradient Boosting (XGBoost), Feedforward neural networks (FNN), and two types of Recurrent neural networks (RNN). All models are applied using different data inputs for daily water temperature prediction in 10 Austrian catchments ranging from 200 km$^2$ to 96000 km$^2$ and exhibiting a wide range of physiographic characteristics. The evaluated input data sets include combinations of daily means of air temperature, runoff, precipitation and global radiation. Bayesian optimization is applied to optimize the hyperparameters of all applied machine learning models. To make the results comparable to previous studies, two widely used benchmark models are applied additionally: linear regression and air2stream.

With a mean root mean squared error (RMSE) of 0.55 $°C$ the tested models could significantly improve water temperature prediction compared to linear regression (1.55 $°C$) and air2stream (0.98 $°C$). In general, the results show a very similar performance of the tested machine learning models, with a median RMSE difference of 0.08 $°C$ between the models. From the 6 tested machine learning models both FNNs and XGBoost performed best in 4 of the 10 catchments. RNNs are the best performing models in the largest catchment, indicating that RNNs are mainly performing well when processes with long-term dependencies are important. Furthermore, a wide range of performance was observed for different hyperparameter sets for the tested models, showing the importance of hyperprameter optimization. Especially the FNN model results showed an extremely large RMSE standard deviation of 1.60 $°C$ due to the chosen hyperparamerters.

This study evaluates different sets of input variables, machine learning models and training characteristics for daily stream water temperature prediction, acting as a basis for future development of regional multi-catchment water temperature prediction models. All preprocessing steps and models are implemented into the open source R package wateRtemp, to provide easy access to these modelling approaches and facilitate further research.

## 1 Introduction

Water temperature in rivers should not be considered only as a physical property, since it is a crucial environmental factor and a substantial key element for water quality and aquatic habitats. In particular, it influences riverine species by governing e.g. metabolism (Álvarez and Nicieza, 2005), distribution (Boisneau et al., 2008), abundance (Wenger et al., 2011), community





composition (Dallas, 2008) and growth (Imholt et al., 2010), thus aquatic organisms have a specific range of river temperature they are able to tolerate (Caissie, 2006). Due to the impact of water temperature on chemical processes (Hannah et al., 2008) and other physical properties such as density, vapour pressure and viscosity (Stevens et al., 1975), stream temperature indirectly

influences key ecosystem processes such as primary production, decomposition and nutrient cycling within rivers (Friberg et al., 2009). These parameters and processes affect the level of dissolved oxygen (Sand-Jensen and Pedersen, 2005) and, of course, have a major influence on water quality (Beaufort et al., 2016).

Besides its ecological importance, river temperature is also of socio-economic interest for electric power and industry (cooling), drinking water production (hygiene, bacterial pollution) and fisheries (fish growth, survival and demographic character-

istics) (Hannah and Garner, 2015). Hence, a changing river temperature can strongly alter the hydro-ecological and socio-economic conditions within the river and its neighboring region. Assessing alterations of this sensitive variable and its drivers is essential for the management of impacts and enabling prevention measurements.

Direct temperature measurements are often scarce and rarely available. For a successful integrated water management, it will be essential to derive how river temperature will be developing in the future, in particular when considering relevant global

change processes (e.g. climate change), but also on shorter time scales. The forecast, for example, of river temperature with a lead time of a few days can substantially improve or even allow the operation of thermal power plants. Two aspects are important: The efficiency of cooling depends on the actual water tempererature. On the other hand, legal constraints regarding maximum allowed river temperatures due do ecological reasons can be exceeded, when warmed up water is directed into the river after the power plant. This is especially relevant during low flow conditions in hot summers. The knowledge of the to be

expected water temperature in the next few days is therefore of advantage. An important step in this context is the development of appropriate modelling concepts to predict river water temperature, to describe thermal regimes and to investigate the thermal development of a river. In Austria there are 210 river water temperature measurement stations available, sometimes with 30+ years of data. This large number of available date in Austria is highly advantageous for developing new modelling concepts. Additionally, a wide range of catchments with different physiographic properties are available, ranging from high-alpine,

glacier dominated catchments to low land rivers, with meandering characteristics.

In the past, various models were developed to investigate thermal heterogeneity at different temporal and spatial scales, the nature of past availability and likely future trends (Laizé et al., 2014; Webb et al., 2008). In general, water temperature in rivers is modelled by process-based models, statistical/machine learning models or a combination of both approaches. Process-based models are representing physical processes controlling river temperature. According to Dugdale et al. (2017), these models are

based on two key steps: First, calculating energy fluxes to or from the river and then determining the temperature change in a second step. Calculating the energy fluxes means to solve the energy balance equation for a river reach by considering the heat fluxes at the air-water and riverbed-water interface (Beaufort et al., 2016). These demanding energy budget components are derived either by field measurements or by approximations (Caissie and Luce, 2017; Dugdale et al., 2017; Webb and Zhang, 1997), highlighting the complexity and parametrisation of this kind of models. Although it is not feasible to monitor these

components over long periods or at all points along a river network and contributing catchments (Johnson et al., 2014), they provide clear benefits: (i) give insights into the drivers of river water temperature, (ii) inform about metrics, which can be used



in larger statistical models and (iii) different impact scenarios (Dugdale et al., 2017). These arguments are also the reasons why data-intensive process-based models are widely used despite of their high complexity.

Statistical and machine learning models are grouped into parametric approaches, including regression (e.g. Mohseni and
Stefan, 1999) and stochastic (e.g. Ahmadi-Nedushan et al., 2007) models, and non-parametric approaches based on computational algorithms like neural networks or k-nearest-neighbours (Benyahya et al., 2007). In contrast to process-based models, statistical models cannot inform about energy transfer mechanisms within a river (Dugdale et al., 2017). Another main concern is that parametric statistical models showed higher prediction performances on weekly, monthly or seasonal time scales in the past (Caissie, 2006) leading to a loss of temporal variation (DeWeber and Wagner, 2014). However, unlike process-based
models, they do not require a large number of input variables, which are unavailable in many cases. Non-parametric statistical models have gained attention in the past few years. Especially machine learning techniques are proofed to be a useful tool in river temperature modelling already (Zhu and Piotrowski, 2020).

For this study we chose a set of state of the art machine learning models that showed promising results for water temperature prediction or in similar time series prediction tasks. The 6 chosen models are: step-wise linear regression, Random Forest, eX-
treme Gradient Boosting (XGBoost), Feedforward neural networks (FNN) and two types of recurrent neural networks (RNN). Step-wise linear regression models combine an iterative variable selection procedure with linear regression models. To our knowledge only one previous study by Neumann et al. (2003) already applied this method for predicting daily maximum river water temperature. The Random Forest model (RF) (Breiman, 2001) is an ensemble learning model that averages the results of multiple regression trees. To the authors knowledge, RF has not been applied for river water temperature prediction yet. Zhu
et al. (2019d) used bootstrap aggregated decision trees, which are similar, but do not include the random variable sampling for splitting the tree nodes, which is an important characteristics of the RF model. XGBoost (Chen and Guestrin, 2016) is also a regression tree based ensemble learning model. However, instead of averaging multiple trees, XGBoost builds complementary trees for prediction. To the authors knowledge, XGBoost has not been applied for river water temperature predictions yet. However, results from short term water quality parameters predictions, which are also including water temperature, show
promising performances (Lu and Ma, 2020; Joslyn, 2018).

Feedforward neural networks (FNN) (White and Rosenblatt, 1963) are the first and simplest type of neural networks. FNNs have already been applied in numerous stream water temperature prediction studies (e.g. Risley et al., 2003; Bélanger et al., 2005; Chenard and Caissie, 2008; Sahoo et al., 2009; Wehrly et al., 2009; McKenna et al., 2010; Westenbroek et al., 2010; Hadzima-Nyarko et al., 2014; DeWeber and Wagner, 2014; Piotrowski et al., 2015; Rabi et al., 2015; Abba et al., 2017; Zhu
et al., 2018; Temizyurek and Dadaser-Celik, 2018; Zhu et al., 2019b, d; Graf et al., 2019; Zhu et al., 2019a). In contrast to FNN, Recurrent Neural Networks (RNN) are networks developed specifically to process sequences of inputs. This is achieved by having internal (hidden) states. While there are many different types of RNNs, we focused on the two most widely known, the Long short-term memory (LSTM) (Hochreiter and Schmidhuber, 1997) and the Gated recurrent unit (GRU) (Cho et al., 2014). To the authors knowledge, RNNs have been used in one study by Stajkowski et al. (2020) in which a LSTM in combination with
a genetic algorithm hyperparameter optimization was used to forecast hourly urban river water temperature. However, LSTMs





have recently been applied in a wide range of hydrological studies and showed promising results for time-series prediction tasks (e.g. Kratzert et al., 2018, 2019; Xiang et al., 2020; Li et al., 2020).

To make findings comparable with other studies investigating this approach, we apply two benchmark models as baseline: linear regression and air2stream (Toffolon and Piccolroaz, 2015). Linear regression models are widely used for river water
temperature studies. While earlier studies used mainly air temperature as a regressor to predict river water temperature (e.g., Smith, 1981; Crisp and Howson, 1982; Mackey and Berrie, 1991; Stefan and Preud'homme, 1993), more recent publications are using a wider range of input variables or some modification to the standard linear regression model (e.g., Caldwell et al., 2013; Li et al., 2014; Segura et al., 2015; Arismendi et al., 2014; Naresh and Rehana, 2017; Jackson et al., 2018; Trinh et al., 2019; Piotrowski and Napiorkowski, 2019). Air2stream is a hybrid model for predicting river water temperature, which combines
a physical based structure with a stochastic parameter calibration. It was already applied in multiple studies over a range of catchments and generally had an improved performance compared to linear regression and other machine learning models (e.g., Piccolroaz et al., 2016; Yang and Peterson, 2017; Piotrowski and Napiorkowski, 2018; Zhu et al., 2019d; Piotrowski and Napiorkowski, 2019; Tavares et al., 2020).

Most studies mainly use air temperature and discharge as inputs for water temperature prediction (e.g. Piccolroaz et al., 2016;
Naresh and Rehana, 2017; Sohrabi et al., 2017), while others use additional information from precipitation (e.g. Caldwell et al., 2013) and/or solar radiation (e.g. Sahoo et al., 2009). Additionally, air temperature can either be included as mean, maximum or minimum daily temperature (e.g. Piotrowski et al., 2015). To further investigate which meteorological and hydrological inputs are important and necessary for water temperature prediction, we here use multiple sets of input data and compare their outcome. Especially knowing how simple models with few data inputs perform in comparison with more complex input
combinations can give insight on how to plan applications of water temperature modelling for a range of purposes.

Machine learning models are generally parameterized by a set of hyperparameters that have to be chosen by the user to maximize performance of the model. Depending on the model, hyperparameters can have a large impact on model performance (Claesen and De Moor, 2015), but are still most often chosen by rules-of-thumb (Hinton et al., 2012; Hsu et al., 2003) or by testing sets of hyperparameters on a predefined grid (Pedregosa et al., 2011). In this study we apply a hyperparameter
optimization using the Bayesian optimization method (Kushner, 1964; Zhilinskas, 1975; Močkus, 1975; Močkus et al., 1978; Močkus, 1989) to minimize the possibility of using unsuitable hyperparameters for the applied models and to investigate the spread in performance depending on the chosen hyperparameters.

This publication presents a thorough investigation of models, input data and model training characteristics for daily stream water temperature prediction. It consists of the application of 6 types of machine learning models on a range of different
catchments using multiple sets of data inputs. This includes novel approaches on input data preparation and hyperparameter optimization. The resulting performance of all models is compared to two widely applied benchmark models to make the presented results comparable. Finally, all methods and models are incorporated into an open source R library to make theses approaches available for researchers and industries.



## 2 Methods

### 2.1 Study sites and data

For this study, 10 catchments with a wide range of physiographic characteristics, human impacts (e.g. hydropower, river regulation) and available observation period length were selected. Including study sites with diverse properties allows for validating the applicability and performance of the introduced modelling approach. The catchments are situated in Austria, Switzerland and Germany, with outlets located in the Austrian Alps or adjacent flatlands. All catchments and gauging stations

are shown in Figure 1 and their main characteristics are summarized in Table 1.

     The gauging stations are operated by the Austrian Hydrographical Service (HZB) and measure discharge ($Q$) in 15 minute intervals and water temperature ($T_w$) in a range of different time intervals (daily mean - 1 minute). The temperature sensors are situated in a way that complete mixing can be assumed, e.g. after a bottom ramp. Consequently, the measured water temperature should reflect the water temperature of the given cross section.

The meteorological data used in this study are daily mean air temperature ($T_a$), daily max air temperature ($T_{max}$), daily min air temperature ($T_{min}$), precipitation sum ($P$) and global radiation ($GL$). $T_a$, $T_{max}$, $T_{min}$ and $P$ were available from the SPARTACUS project (Hiebl and Frei, 2016, 2018) on a 1x1 km grid from 1961 onward. The SPARTACUS data was generated by using observations and external drift kriging to create continuous maps. $GL$ data were available from the INCA analysis (Integrated Nowcasting through Comprehensive Analysis) (Haiden et al., 2011, 2014) from 2007 onward. The INCA analysis

used numerical weather simulations in combination with observations and topographic information to provide meteorological analysis and nowcasting fields of several meteorological parameters on a 1x1 km grid in 15-60 minute time steps. For the presented study, the 15 minute INCA $GL$ analysis fields were aggregated to daily means. The catchment means of all variables are shown in Table 1. By using high resolution spatially distributed meteorological data as basis for our inputs, we aim to better represent the main drivers of water temperature changes in the catchments. Similar data sets are available for other parts of

the world, e.g. globally (Hersbach et al., 2020), for North America (Thornton et al., 2014; Werner et al., 2019), for Europe (Brinckmann et al., 2016; Razafimaharo et al., 2020) and for China (He et al., 2020).

### 2.2 Data preprocessing

The applied data preprocessing consists of aggregation of gridded data, feature engineering (i.e. deriving new features from existing inputs) and splitting the data into multiple sets of input variables. Since river water temperature is largely controlled

by processes within the catchment, variables with an integral effect on water temperature over the catchment (i.e. $T_a$, $T_{max}$, $T_{min}$, $P$ and $GL$) are aggregated to catchment means.

     Computing additional features from a given data set (i.e. feature engineering) and therefore having additional data representation, can significantly improve the performance of machine learning models (Bengio et al., 2013). Previous studies have shown that especially time information is important for water temperature prediction. This includes time expressed as day of

the year (e.g., Hadzima-Nyarko et al., 2014; Li et al., 2014; Jackson et al., 2018; Zhu et al., 2018, 2019c, d), the content of the Gregorian calendar (i.e. year, month, day) (Zhu et al., 2019b), or expressed as the declination of the sun (Piotrowski et al.,



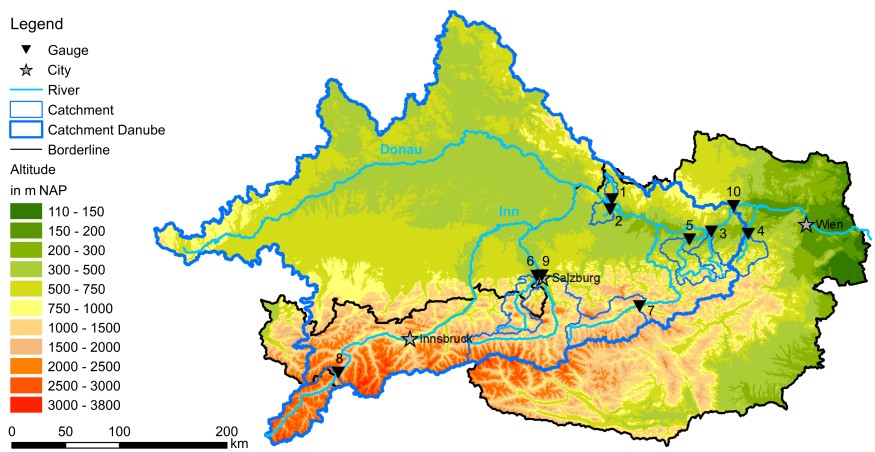

**Figure 1.** Study sites in Austria, Germany and Switzerland. All gauging station IDs refer to the IDs in Table 1.

**Table 1.** Overview of study catchment characteristics, including means of meteorological values of catchment means, catchment areas, available data time periods and number of years with data. IDs are referring to the IDs used in Figure 1.

| ID | Catchment | Gauging Station | Time period | Years | Area $(km^2)$ | $T_w$ $(°C)$ | $Q$ $(m^3/s)$ | $T_a$ $(°C)$ | $P$ $(mm)$ | $GL$ $(W/m^2)$ |
|----|-----------|-----------------|-------------|-------|---------------|--------------|---------------|--------------|------------|----------------|
| 1  | Kleine Mühl | Obermühl      | 2002-2015   | 14.0  | 200.2         | 8.87         | 3.12          | 9            | 2.73       | 135            |
| 2  | Aschach   | Kropfmühle      | 2004-2015   | 11.9  | 312.2         | 10.78        | 3.80          | 10           | 2.50       | 136            |
| 3  | Erlauf    | Niederndorf     | 1980-2015   | 35.3  | 604.9         | 9.42         | 15.27         | 8            | 3.59       | 127            |
| 4  | Traisen   | Windpassing     | 1998-2015   | 17.7  | 733.3         | 9.83         | 14.88         | 8            | 3.33       | 131            |
| 5  | Ybbs      | Greimpersdorf   | 1981-2015   | 34.7  | 1 116.6       | 9.87         | 31.50         | 8            | 3.77       | 127            |
| 6  | Saalach   | Siezenheim      | 2000-2015   | 16.0  | 1 139.1       | 8.50         | 39.04         | 7            | 4.60       | 135            |
| 7  | Enns      | Liezen          | 2006-2015   | 10.0  | 2 116.2       | 1.19         | 67.56         | 6            | 3.60       | 137            |
| 8  | Inn       | Kajetansbrücke  | 1997-2015   | 18.8  | 2 162.0       | 6.00         | 59.26         | 0            | 2.56       | 153            |
| 9  | Salzach   | Salzburg        | 1977-2015   | 39.0  | 4 425.7       | 7.63         | 178.11        | 5            | 4.16       | 136            |
| 10 | Donau     | Kienstock       | 2005-2015   | 11.0  | 95 970.0      | 10.77        | 1 798.31      | 10           | 2.13       | 131            |

2015), which is a function of the day of the year. Nevertheless, using cyclical features like day of the year as an integer variable will most likely reduce model performance, since the days 1 and 365 are as close together as 1 and 2. To translate time information into a more suitable format we chose to transform months and days of months into trapezoidal fuzzy sets, called fuzzy

months. Similar to dummy encoding, the values of a fuzzy month are between 0 and 1. They are equal 1 on the 15th of the corresponding month and linearly decreasing each day until they are zero on the 15th day of the previous and following month.





Therefore, the values of two adjacent months will be around 0.5 at the turn of the month. By encoding the categorical variable "month" into these 12 new fuzzy variables, it should be possible to represent time of the year influence more smoothly, as no jumps in monthly influence are possible. Initial test showed that the advantage of this representation exceeds the disadvantage
of using 12 variables instead of one or two. A similar approach for encoding time variables was already applied by Shank et al. (2008).

Beside time variables, a previous study by Webb et al. (2003) showed that lag information are significantly associated with water temperature and can improve model performance. Therefore, to allow for information of previous days to be used by the models, the lags of all variables for the 4 previous days are computed and used as additional features.

Using these input variables six experiments with different sets of inputs considering different levels of data availability are defined. The variable compositions of all experiments are shown in Table 2. All features include 4 lags and each experiment also includes fuzzy moths as inputs. Experiment 0 (Tmean) acts as another simple benchmark in which only mean water temperature and fuzzy months are used for predictions. Experiment 1 (T) will be able to show the benefit of including $T_{max}$ and $T_{min}$. Experiments 2-4 consists of combinations of experiment 1 with precipitation and discharge data. Experiments 5-6 include combinations with $GL$ and therefore include only data of the time period 2007-2015 in which $GL$ data was available.

**Table 2.** Overview of available meteorological and hydrological variables and the composition of the different input data set experiments. If an input variable is used in a data set the lags for the 4 previous days are included as well. Additional to the shown variables, all experiments use fuzzy months as input.

| Experiment | $T_a$ | $T_{max}$ | $T_{min}$ | $P$ | $Q$ | $GL$ |
|---|---|---|---|---|---|---|
| 0 (Ta) | **X** | | | | | |
| 1 (T) | **X** | **X** | **X** | | | |
| 2 (TP) | **X** | **X** | **X** | **X** | | |
| 3 (TQ) | **X** | **X** | **X** | | **X** | |
| 4 (TQP) | **X** | **X** | **X** | **X** | **X** | |
| 5 (TPGL) | **X** | **X** | **X** | **X** | | **X** |
| 6 (TQPGL) | **X** | **X** | **X** | **X** | **X** | **X** |


## 2.3 Benchmark models

Two widely applied models for stream water temperature prediction are used as benchmark for all models tested in this study: multiple linear regression models (LM) and air2stream (Toffolon and Piccolroaz, 2015). By including these two models, it will be possible to compare this study's results to a wider range of previous studies, which were investigating models for stream
water temperature prediction.



### 2.3.1 Linear regression

Linear regression models are widely used for river water temperature studies. Earlier studies used mainly air temperature as a regressor to predict river water temperature (e.g., Smith, 1981; Crisp and Howson, 1982; Mackey and Berrie, 1991; Stefan and Preud'homme, 1993). More recent publications are using a wider range of input variables or some modification to the standard linear regression model (e.g., Caldwell et al., 2013; Li et al., 2014; Segura et al., 2015; Arismendi et al., 2014; Naresh and Rehana, 2017; Jackson et al., 2018; Trinh et al., 2019; Piotrowski and Napiorkowski, 2019).

The ordinary least square linear regression model is defined as:

$$\mathbf{Y} = \beta\mathbf{X} + \epsilon \tag{1}$$

where $\mathbf{Y}$ denotes the vector of the dependent variable (river water temperature), $\mathbf{X}$ denotes the matrix of independent variables (e.g. daily mean air temperature, global radiation), $\beta$ denotes the vector of model coefficients and $\epsilon$ denotes the error term. $\epsilon$ is assumed to be normal distributed with a diagonal covariance matrix. The estimates for the model coefficients and the independent variable, which minimize the sum of squared errors are given by

$$\hat{\mathbf{Y}} = \hat{\beta}\mathbf{X} \tag{2}$$

$$\hat{\beta} = (\mathbf{X}'\mathbf{X})^{-1}\mathbf{X}'\mathbf{Y} \tag{3}$$

where $\hat{\mathbf{Y}}$, and $\hat{\beta}$ represent estimated values. The linear regression model applied in this study includes an intercept and the variables $T_a$ and $Q$ as independent variables to predict $T_w$.

### 2.3.2 air2stream

Air2stream (Toffolon and Piccolroaz, 2015) is a hybrid model for predicting river water temperature, which combines a physical based structure with a stochastic parameter calibration. It was already applied in multiple studies over a range of catchments and generally had an improved performance compared to linear regression models (e.g., Piccolroaz et al., 2016; Yang and Peterson, 2017; Piotrowski and Napiorkowski, 2018; Zhu et al., 2019d; Piotrowski and Napiorkowski, 2019; Tavares et al., 2020). Air2stream uses the inputs $T_a$ and $Q$ and was derived from simplified physical relationships expressed as ordinary differential equations for heat budged processes. Due to this simplification it may be applied like a data driven model, which depends on parameter calibration. The 8-parameter version of air2stream is defined as

$$\frac{dT_w}{dt} = \frac{1}{\theta^{a_4}}[a_1 + a_2 T_a - a_3 T_w + \theta(a_5 + a_6 cos(2\pi(\frac{t}{t_y} - a_7)) - a_8 T_w)] \tag{4}$$

where t is the time in days, $t_y$ is the number of days per year, $\bar{Q}$ the mean discharge, $\theta = Q/\bar{Q}$ is the dimensionless discharge and $a_{1,...,8}$ the model parameters. This differential equation is numerically integrated at each time step using the Crank-Nicolson numerical scheme (Crank and Nicolson, 1947) and the model parameters are calibrated using the Particle Swarm Optimization (Kennedy and Eberhart, 1995).





## 2.4 Machine Learning Models

In this study we compare 6 different machine learning models: step-wise linear regression (step-LM), Random Forest (RF), eXtreme Gradient Boosting (XGBoost), Feedfoward neural networks (FNN) and two Recurrent neural networks (RNN) - the a Long short-term network (RNN-LSTM) and the Gated recurrent unit (RNN-GRU). An overview and simple depiction of the models is shown in Figure 2.

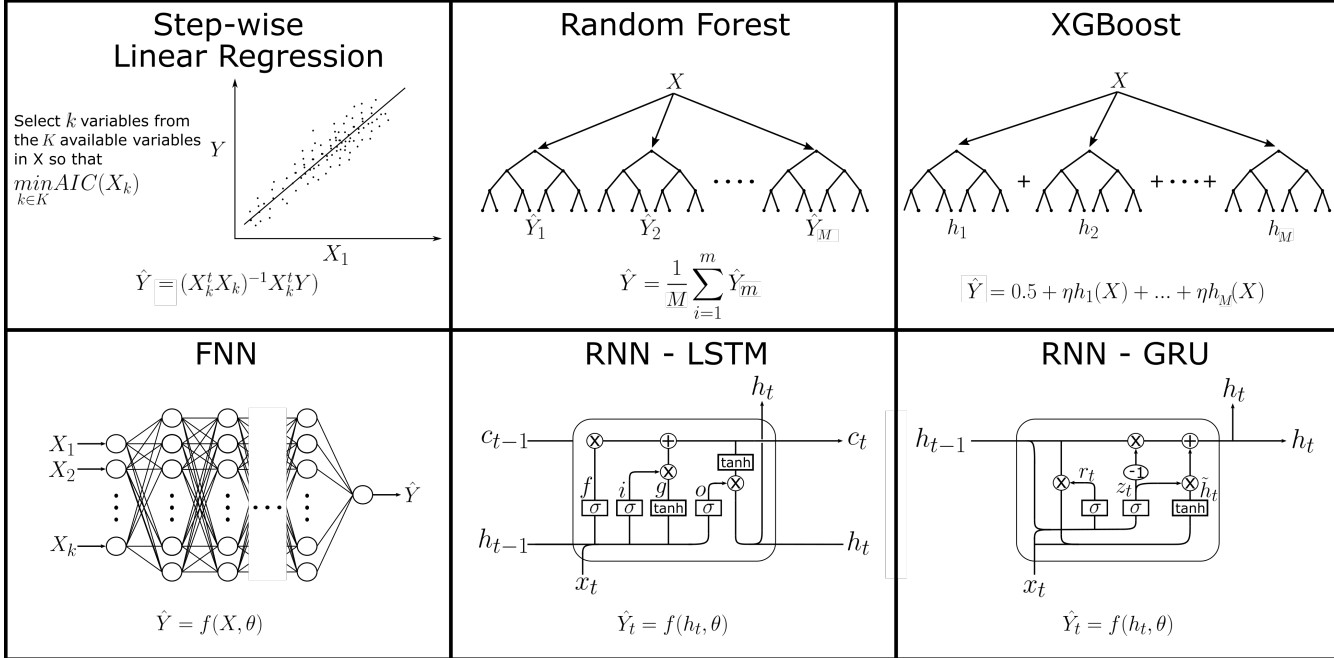

**Figure 2.** Overview of the applied models with $\hat{Y}$ denoting estimated water temperatures and X the matrix of observed variables. $\hat{Y}_{1,...,M}$ are the predictions from individual RF trees. $h_{1,...,M}$ are the predicted residuals from individual XGBoost trees. $f(X, \theta)$ denotes a mapping from a FNN with the parameters $\theta$. For a given time step, $h_t$ denotes the hidden internal state of a RNN cell and $c_t$ the internal cell state of a LSTM cell and $f(h_t, \theta)$ the mapping from a RNN with the parameters $\theta$. RNNs consist of a cascade of cells, each feeding their internal states into the next cell, finally resulting in a single feedforward layer estimating $\hat{Y}$ from $h_t$.

## 2.4.1 Step-wise linear regression

Step-wise linear regression models combine an iterative variable selection procedure with linear regression models. The step-wise variable selection starts at an initial model (e.g. all variables) and removes or adds at each iteration based on a prespecified criterion. We applied the step-wise variable selection starting with an initial model including all variables and using the Akaike Information Criterium (AIC) (Akaike H., 1973). The AIC for a linear regression model is given by:

$$AIC = n \times ln(\frac{\sum_n^{i=1}(Y_i - \hat{Y}_i)^2}{n} + 2k \qquad (5)$$





where n is the number of samples, $ln()$ the natural logarithm, $Y$ and $\hat{Y}$ the observed and predicted water temperatures and k the number of selected input variables. The step-wise variable selection is iteratively applied until AIC is at a minimum. Additional to the variables given in section 2.2, interactions terms between $T_a$, $Q$, $GL$ and $P$ are included.

### 2.4.2 Random Forest

The Random Forest model (RF) (Breiman, 2001) is an ensemble learning model based on the idea of bagging (bootstrap aggregating) (Breiman, 1996). Bagging predictors average multiple model predictions, where each model is trained on a boot-strapped sample instead of the full observed sample. This randomness introduced by bootstrapping increases the models ability to generalize and to produce stable prediction results.

RF models are bagging predictors which use classification and regression trees (CARTs) as a base learner. RF CARTs
recursively apply binary splits to the data to minimize entropy in the tree nodes. This is done until each node reaches a minimum node size or a previously defined maximum tree depth is reached. Breiman (2001) showed that adding further randomness to the bagging method improves prediction accuracy. In random forests this is achieved by only selecting a random subset of available variables for the split at each node. The estimate for the independent variable is given by

$$\hat{\mathbf{Y}} = \frac{1}{M} \sum_{m=1}^{M} f_i(\mathbf{X}) \tag{6}$$

where $f_m$ denotes a single fitted CART, $M$ the number of used CARTs, $\mathbf{X}$ the matrix of regressors and $\hat{\mathbf{Y}}$ the vector of estimated water temperature. A simplified depiction of the RF algorithm is shown in Figure 2. RF has 2 important hyperparameters: the number of predictors sampled at each node (mtry) and the minimum size of nodes (min node size). The number of trees was chosen to be constant with 500 trees.

### 2.4.3 XGBoost

XGBoost (Chen and Guestrin, 2016) is a tree boosting algorithm that was developed based on the already existing concept of boosting, which was further enhanced to increase efficiency, scalability and reduced overfitting. Similar to bagging, boosting methods combine the prediction of an ensemble of weak learner to improve prediction accuracy. However, while bagging ensemble members are trained in parallel, boosting iteratively trains new ensemble members and adds them to the existing ensemble. Boosting was first introduced by Schapire (1990) and then widely applied after the introduction of the Adaboost
algorithm (Freund and Schapire, 1995). Friedman (2001) further enhanced boosting by adding gradient decent optimization for the boosting iterations. This resulted in the development of gradient tree boosting (Friedman, 2002), which uses CART as weak learners.

XGBoost is an implementation of gradient tree boosting with further enhancements in the form of added stochasticity and regularization. The XGBoost estimated for the independent variable is given by

$$\hat{\mathbf{Y}} = 0.5 + \sum_{m=1}^{M} \eta f_m(\mathbf{X}) \tag{7}$$





where $f_1, ... f_M$ is a sequence of CARTs, $\eta \in [0, 1]$ is the learning rate, $M$ is the number of used CARTs, $\mathbf{X}$ is the matrix of input features and $\hat{\mathbf{Y}}$ is the vector of estimated water temperatures. The m-th tree is trained to predict the residuals of a model of the form given in equation 7, which uses the previous m-1 CARTs. The loss function used to train each tree includes a regularization term to prevent overfitting. Additionally, overfitting is reduced by only allowing a random subset of samples and variables, be used for constructing trees and tree nodes at each iteration. A simplified depiction of the XGBoost algorithm is shown in Figure 2.

XGBoost has multiple important hyperparameters that have to be chosen before fitting the model: the maximum number of iterations (nrounds), the learning rate ($\eta$), the maximum depth of a tree (max depth), the minimum sum of instance weight needed in a child (min node size), the ratio of random subsamples used for growing a tree (subsample) and the random fraction of variables used for growing a tree (colsample bytree).

### 2.4.4 Feedforward Neural Network

Feedforward neural networks (FNN) (White and Rosenblatt, 1963) are the first and simplest type of neural networks. FNNs consists of multiple layers of nodes, where each node is connected to all nodes of the previous and following layer. A node applies a linear and a non-linear (activation-) function to its input to produce an output. The general structure of a FNN is shown in Figure 2.

Piotrowski et al. (2020) showed that adding dropout (Hinton et al., 2012; Srivastava et al., 2014; Baldi and Sadowski, 2014) to FNNs for stream water temperature prediction improved performance of single layer FNNs. Dropout refers to randomly dropping nodes from a layer during training, which can prevent overfitting and potentially improve generalization. We added a dropout to every FNN layer and defined the dropout rate as a hyperparameter.

While the parameters ($\theta$) of the linear function get optimized using backpropagation (Rumelhart et al., 1986), FNN have multiple hyperparameters that need to be predefined before training. These hyperparameters include the activation functions, the number of layers, the number of nodes per layer and the dropout ratio. After initial tests, in which a large set of different activation function were applied, we chose the Scaled Exponential Linear Unit (SELU) activation function (Klambauer et al., 2017) for all nodes in the network. SELU includes a normalization, which enhances convergence and avoids both vanishing and exploding gradients during backpropagation. The other hyperparameters are optimized as described in section 2.5.

The here presented hyperparameter optimization approach differs from previous studies, which generally assume a set of fixed number of layers and/or nodes per layer that were derived by a trial and error approach (e.g., Bélanger et al., 2005; Hadzima-Nyarko et al., 2014; Piotrowski et al., 2015; Zhu et al., 2018, 2019d).

### 2.4.5 Recurrent Neural Networks

In contrast to FNN, Recurrent Neural Networks (RNN) are able to process sequences of inputs. This is achieved by having internal (hidden) states. While there are many different types of RNNs, we focused on the two most widely known, the Long short-term memory (LSTM) (Hochreiter and Schmidhuber, 1997) and the Gated recurrent unit (GRU) (Cho et al., 2014). Each layer of an RNN consists of a sequence of cells that share a common set of weights. The cells of both LSTM and GRU are





shown in Figure 2 and are described in the next sections. A single RNN cell consists of multiple gates, which refers to the

nodes of a cell where non-linear transformations are applied to the inputs and states. Each RNN contains a FNN layer with a single node at its end, which is used to compute the predicted values from the hidden states of the last time step ($\mathbf{h_T}$). Both types of RNNs have the same set of hyperparameters that need to be specified before training the model: the number of used RNN layers, the number of units per layer, the numbers of timesteps, the dropout ratio, and the batch size.

Due to their internal states and the usage of multiple time steps for prediction, it can be assumed that RNNs do not need time

information (here in form of fuzzy months) for predicting water temperature data. To test this assumption both RNN variants are also trained without fuzzy months to check the influence of these additional variables on model performance. Being able to achieve equally good results without fuzzy months, would reduce training time considerably due to decreasing the input data by 12 dimensions (columns).

**Long short-term memory cells**

Given a sequence of inputs for $T$ time steps $x_1, ..., x_T$, where each $x_t \in \mathbb{R}^d$ is a vector of $d$ input features, the forward pass of a single LSTM cell with $h$ hidden units is given by following equations:

$$\mathbf{f_t} = \sigma(\mathbf{W_f x_t} + \mathbf{U_f h_{t-1}} + \mathbf{b_f}), \tag{8}$$

$$\mathbf{i_t} = \sigma(\mathbf{W_i x_t} + \mathbf{U_i h_{t-1}} + \mathbf{b_i}), \tag{9}$$

$$\mathbf{o_t} = \sigma(\mathbf{W_o x_t} + \mathbf{U_o h_{t-1}} + \mathbf{b_o}), \tag{10}$$

$$\mathbf{\tilde{c}_t} = tanh(\mathbf{W_g x_t} + \mathbf{U_g h_{t-1}} + \mathbf{b_g}), \tag{11}$$

$$\mathbf{c_t} = \mathbf{f_t} \odot \mathbf{c_{t-1}} + \mathbf{i_t} \odot \mathbf{\tilde{c}_t}, \tag{12}$$

$$\mathbf{h_t} = \mathbf{o_t} \odot tanh(\mathbf{c_t}) \tag{13}$$

where $\mathbf{f_t}$, $\mathbf{i_t}$, and $\mathbf{o_t} \in \mathbb{R}^h$ are the forget gate, input gate and output gate, $\mathbf{\tilde{c}_t} \in \mathbb{R}^h$ is the cell input activation, $\mathbf{h_t} \in \mathbb{R}^h$ is the hidden state, $\mathbf{c_t} \in \mathbb{R}^h$ is the cell state and all $\mathbf{W} \in \mathbb{R}^{h \times d}$, $\mathbf{U} \in \mathbb{R}^{h \times h}$ and $\mathbf{b} \in \mathbb{R}^h$ are trainable weights. $\sigma$ is the sigmoid

function and $tanh$ the hyperbolic tangent function and $\odot$ is element-wise multiplication.

The hidden state ($h_t$) is computed from the current input ($x_t$) and the previous hidden state ($h_{t-1}$). The amount of information that is passed through the current cell is regulated by the input gate ($i_t$) and the forget gate ($f_t$). The cell state ($c_t$) regulates how much memory will be stored in the hidden state ($h_t$). The output gate ($o_t$) controls how much information is passed to the next cell.

**Gated Recurrent Units cells**

The GRU cell is similar to a LSTM cell, but much simpler. It combines the forget and input gate into a single update gate and also merges the cell state and the hidden state. Given a sequence of inputs for $T$ time steps $x_1, ..., x_T$, where each $x_t \in \mathbb{R}^d$





is a vector of $d$ input features, the forward pass of a single GRU cell with $h$ hidden units is given by following equations:

$$\mathbf{z_t} = \sigma(\mathbf{W_z x_t} + \mathbf{U_z h_{t-1}} + \mathbf{b_z}) \tag{14}$$

$$\mathbf{r_t} = \sigma(\mathbf{W_r x_t} + \mathbf{U_r h_{t-1}} + \mathbf{b_r}) \tag{15}$$

$$\mathbf{\hat{h}_t} = \tanh(\mathbf{W_h x_t}) + \mathbf{U_h}(\mathbf{r_t} \odot \mathbf{h_{t-1}} + \mathbf{b_h}) \tag{16}$$

$$\mathbf{h_t} = (1 - \mathbf{z_t}) \odot \mathbf{h_{t-1}} + \mathbf{z_t} \odot \mathbf{\hat{h}_t} \tag{17}$$

where $\mathbf{z_t} \in \mathbb{R}^h$ is the update gate, $\mathbf{r_t} \in \mathbb{R}^h$ is the reset gate, $\hat{\mathbf{h}}_t \in \mathbb{R}^h$ is the candidate activation, $\mathbf{h_t} \in \mathbb{R}^d$ is the output and all $\mathbf{W} \in \mathbb{R}^{h \times d}$, $\mathbf{U} \in \mathbb{R}^{h \times h}$ and $\mathbf{b} \in \mathbb{R}^h$ are trainable weights.

The reset gate ($r_t$) determines how much information from the previous state will be forgotten when computing the candidate activation ($\hat{\mathbf{h}}_t$). The update gate is the amount of information used from the candidate activation ($\hat{\mathbf{h}}_t$) for computing the current output $h_t$.

## 2.5    Bayesian hyperparameter optimization

Choosing adequate hyperparameters for a machine learning model can have a large impact on its performance. Therefore, it
is necessary to apply some sort of optimization procedure. While it might be possible to apply a grid search over the range of all possible parameter value combinations for a small set of hyperparameters, it is usually not feasible due to available computational resources. For that reason, we chose to optimize the hyperparameters of nearly all machine learning models in this study with the bayesian optimization method. Only Random forest with 3 hyperparameters is optimized using a grid search. Step-wise linear regression does not have hyperparameters that need optimization.

Bayesian optimization is a global optimization method for black-box functions (i.e. lacks known structure and is derivative-free) that is often applied in cases where the objective function is computational expensive to evaluate. It originates from work by Kushner (1964), Zhilinskas (1975), Močkus (1975), Močkus et al. (1978), Močkus (1989) and was later popularized by Jones et al. (1998). It got especially well known for being suitable for optimizing machine learning hyperparameters after a study by Snoek et al. (2012).

Bayesian optimization consists of two parts: a method for statistical inference and an acquisition function for deciding the next sample point. The method for statistical inference is usually a Gaussian process (GP) which provides an estimated posterior distribution at each iteration that is an estimate for the function that should be optimized. The acquisition function is used to find the next point to evaluate during each optimization step and was chosen to be the Upper Confidence Bound (UCB) (Srinivas et al., 2009) in this study. In summary, bayesian optimization constructs a surrogate model at each iteration during
optimization to choose a suitable next point. The hyperparameters of all optimized models and their chosen bounds are given in the Appendix A.





## 2.6 Evaluation metrics

The objective function for all models and the hyperparameter optimization is the mean squared error (MSE):

$$MSE = \frac{1}{n}\sum_{i=1}^{n}(y_i - \hat{y}_i)^2 \tag{18}$$

where $n$ is the number of samples (days) and $y_i$ the observed and $\hat{y}_i$ the predicted water temperatures. For comparing the performance of different models the root mean squared error (RMSE) and the mean absolute error MAE are used:

$$RMSE = \sqrt{MSE} \tag{19}$$

$$MAE = \frac{1}{n}\sum_{i=1}^{n}|y_i - \hat{y}_i| \tag{20}$$

## 2.7 Experimental setup

To be able to objectively compare all applied models, the available data sets are split into two parts: the first 80% of the time series were used for training/validation and the last 20% were used for testing. We deliberately did not choose a random split, because predicting water temperatures for a future time period is a more adequate test for models. This is especially relevant for water temperature, which is characterized by non-stationarity due to climate change (Van Vliet et al., 2013). The training/validation and test time series are compared to assess the difference of water temperature distribution of all catchments.

The step-wise linear regression model, RF and XGBoost are optimized using cross-validation (CV). Two kinds of CV are applied: a 5 times repeated 10-fold CV and a time series CV. While the 10-fold CV splits the data randomly, the time series CV gradually adds data to an initial part of the time series while evaluating the performance of each step. The time-series CV started with an initial window of 730 days for training the following 90 days for validation. The training set is increased by 90 days at each different cross validation set until the full time series except for the last 90 days was used. Therefore, instead of

10 folds, the number of folds for the time-series CV depends on the time series length.

Due to computational and time constrains hyperparameter optimization for all neural networks was done by using a 60%/20% of the training/validation data. This allows model validation performance estimation by training a model once, while a 5 times repeated 10-fold CV would require training a model 50 times. This is the standard way of training neural networks for real-world applications.

Bayesian hyperparameter optimization consists of 20 random parameter samples and 40 iterations of optimization. The data inputs for all neural networks were standardized by subtracting the mean and dividing by the standard deviation of the training data. The optimized neural networks hyperparameter sets are used to create 5 independently trained models, from which an ensemble for prediction is created by taking the average of all 5 prediction results. Using ensembles of networks is a way to significantly increase a neural networks ability to generalize and is an often applied approach which was first introduced by the

work of Hansen and Salamon (1990). In addition, early stopping with patience = 5 was applied to all neural networks to avoid overfitting.





The best performing model for each model type and experiment is chosen using the validation RMSE. Test RMSE and MAE
results are only compared after choosing the models with minimum validation RMSE. Consequently, it might be possible that
some models have a superior test performance, but are not chosen as the best performing model for a specific model type and/or
experiment. This should reflect a real-world application, where test data acts as a previously unknown future time series.

## 2.8   Statistical tests

The Kruskal-Wallis test (Kruskal and Wallis, 1952) was used to test for differences in overall model performances, different
training/model characteristics and different data inputs. Dunn's test for multiple comparison (Dunn, 1964) was used for pair-
wise comparisons between model performances. To investigate the association of model types, experiments and catchments
with test RMSE an ordinary least square linear regression model was used. Level of significance was set to $p = 0.05$ for all
statistical tests.

## 2.9   Open-source R package

All preprocessing steps and models were implemented in the open source R package *wateRtemp*, which is available under
github.com/MoritzFeigl/wateRtemp or from Feigl (2021a). This provides easily applicable modelling tools for the water tem-
perature community and allows all results of this study to be replicated. All programming was done in R (R Core Team, 2020),
where the model development relied heavily on Caret (Kuhn, 2020), xgboost (Chen et al., 2020) and TensorFlow (Allaire and
Tang, 2020), and the visualizations on ggplot2 (Wickham, 2016).

# 3   Results

## 3.1   Time period characteristics

Due to climate change induced steadily increasing air temperatures, also water temperature is not a stationary variable and
exhibits warming trends. This is clearly visible when comparing the change in number of extreme warm days and the increase
of mean water temperature in all catchments with time. For this we compared the training/validation and test time data in each
catchment. Since test data consists of the last 20% of the overall data, the exact length of these time series is depending on the
catchment where the test time consists of years in 2008-2015. We can observe an increase of 138% of the median number of
days with water temperature above the 90% quantile between training/validation and test time period in all catchments. This
increase ranges from 69%, or from 32 days to 54 days, in the Donau catchment and up to 285%, or from 26 days to 100 days,
in the Salzach catchment. This change is even more pronounced when comparing the last year of test data (2015) to all other
available years, where the median number of days with water temperatures above the 90% quantile (computed for the overall
timeseries) of all catchments increases by 273%. Figure 3 shows the corresponding boxplots of days with stream temperature
above the 90% quantile for each catchment in training/validation and in test time period. A similar pattern can be observed in
the changes of mean yearly stream temperatures. The median increase of mean yearly water temperature of all catchments is





0.48 $°C$ when comparing training/validation with test time period and 0.77 $°C$ when comparing the last year of the test period (2015) with all other years. Since the test period is, as shown here regarding extremes, different from the training/validation period, the models are also, at least to some extent, tested on how they perform under instationary conditions. This is a test, where environmental models often fail (e.g. Kling et al., 2015).

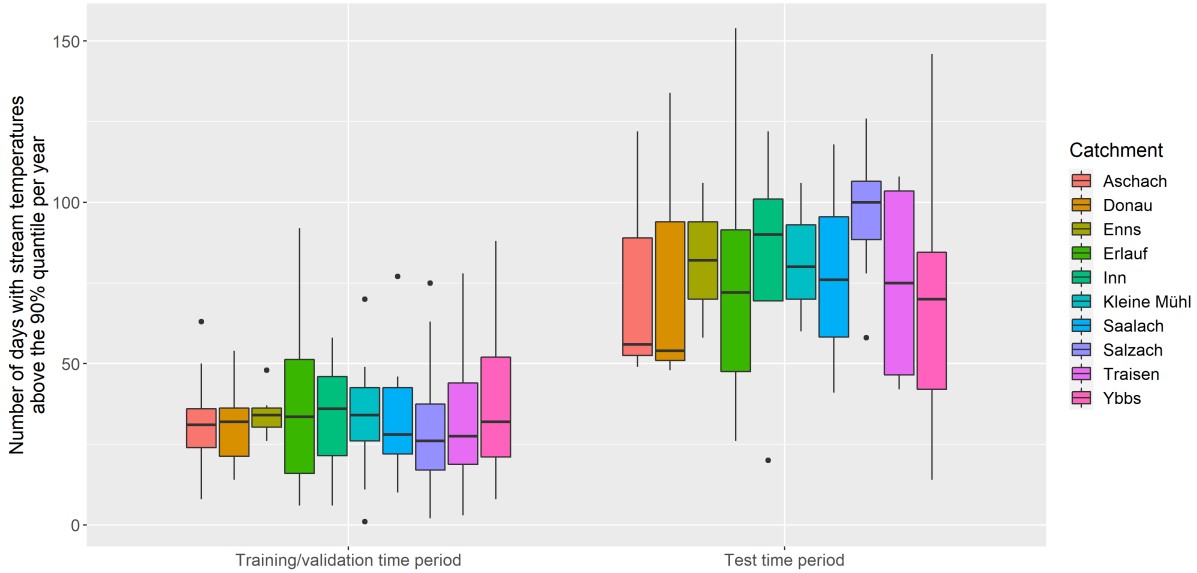

**Figure 3.** Boxplots showing the distribution of numbers of days with stream temperatures above the 90% quantile per year for all study catchments for the training/validation and the test time period, where the test time period consists of the last 20% of data in each catchment. The 90% quantile values were estimated using the full time series for each catchment.


## 3.2 Overall performance comparison

Table 3 gives an overview of the best applied machine learning models and the two benchmark models LM and air2stream. The mean test RMSE of LM is 1.55 $°C$ with an overall range of [1.25, 2.15]$°C$, while air2stream has a mean test RMSE of 0.98 $°C$ with an overall range of [0.74, 1.17] $°C$. The performance results for each catchment show that air2stream always

outperformed LM and consequently results in a significant lower test RMSE ($p < 0.001$). The mean test RMSE of the best machine learning models per catchment is 0.55 $°C$ with an overall range of [0.42, 0.82] $°C$ and always outperformed the air2stream benchmark. Based on the RMSE means, the highest performing ML model is 64% and 43% better, compared to LM and air2stream. This results in a significantly lower test RMSE of the tested machine learning models compared to the air2stream benchmark ($p < 0.001$).

Both XGBoost and FNN were found to be the best performing model in 4 of 10 analysed catchments each. RF was the best performing model in the Salzach catchment and RNN-LSTM in the Donau catchment. Step-LM and RNN-GRU did not





outperform the other models in any of the study catchments. Experiment 3, which only includes air temperature and discharge input features, resulted in the best performing model in four catchments. Experiment 6, which included all available input features, also produced the best performing model in four catchments. Experiment 4, which includes air temperature, discharge and precipitation input features, performed best in two catchments.

**Table 3.** Overview of model performance of the best machine learning model for each catchment and the two reference models. The best performing model results in each catchments are shown in bold font. The best machine learning model for each catchment was chosen by comparing validation RMSE values, while test RMSE and test MAE values were never part of any selection or training procedure. The shown values all refer to the test time period.

| Catchment | Model | Experiment | Best ML model results RMSE ($°C$) | MAE ($°C$) | LM RMSE ($°C$) | MAE ($°C$ | air2stream RMSE ($°C$) | MAE ($°C$) |
|---|---|---|---|---|---|---|---|---|
| Kleine Mühl | XGBoost | 4 (TQP) | **0.740** | **0.578** | 1.744 | 1.377 | 0.908 | 0.714 |
| Aschach | XGBoost | 6 (TQPGL) | **0.815** | **0.675** | 1.777 | 1.408 | 1.147 | 0.882 |
| Erlauf | XGBoost | 6 (TQPGL) | **0.530** | **0.419** | 1.354 | 1.057 | 0.911 | 0.726 |
| Traisen | FNN | 3 (TQ) | **0.526** | **0.392** | 1.254 | 0.970 | 0.948 | 0.747 |
| Ybbs | RF | 3 (TQ) | **0.576** | **0.454** | 1.787 | 1.415 | 0.948 | 0.756 |
| Saalach | XGBoost | 6 (TQPGL) | **0.527** | **0.420** | 1.297 | 1.062 | 0.802 | 0.646 |
| Enns | FNN | 6 (TQPGL) | **0.454** | **0.347** | 1.425 | 1.166 | 1.168 | 0.671 |
| Inn | FNN | 3 (TQ) | **0.422** | **0.329** | 1.376 | 0.098 | 1.097 | 0.949 |
| Salzach | FNN | 4 (TQP) | **0.430** | **0.338** | 1.327 | 1.077 | 0.743 | 0.595 |
| Donau | RNN-LSTM | 3 (TQ) | **0.521** | **0.415** | 2.145 | 1.721 | 1.099 | 0.910 |
| | Mean: | | **0.554** | **0.437** | 1.549 | 1.235 | 0.977 | 0.760 |

Figure 4 shows the results of all models, catchments and experiment combinations. The boxplots in Figure 4a show the range of model performances depending on the model type. Kruskal-Wallis-Test results show no significant difference ($p = 0.11$) of the test RMSE of different model types. Figure 4b shows boxplots of model performance for all experiments. Kruskal-Wallis-Test results show a highly significant difference of test RMSE of the different experiments ($p < 10^{-14}$). The results in Figure 4b show an increase in median performance with increasing number of input features until experiment 4 (TQP). When adding global radiation as additional input parameter, the median performance is not further increasing. This could be explained by a reduced time series length of experiment 5 (TPGL) and 6 (TQPGL), since global radiaton was only available from 2007 on. A comparison between experiments with equal time series lengths (experiments 0-4 and experiments 5-6) also indicate that runoff information improves the modelling performance.

Figure 4c illustrates the RMSE performance results for each catchment shown as boxplots. A corresponding figure of the MAE results is shown in A1. The boxplots are overlayed with scatterplot points adding an overview of the individual performance of each model and experiment combination. To account for a better visibility, the scatterplot points are shifted in





horizontal direction randomly. The difference in performance between catchments is clearly visible and ranges from a median RMSE of around 0.93 $°C$ in catchments Kleine Mühl and Aschach down to a median RMSE of 0.58 $°C$ in the Inn catchment.

Figure 4c also includes the air2stream benchmark performance shown as a grey line for each catchment. Nearly all tested experiments and model combinations showed improved performance compared to the air2stream benchmark. Only in 5 catchments we could observe models in combination with experiments 0, 1, 5 and one time with experiment 6 that predicted worse than air2stream. This are surprisingly few models considering the fact that both experiment 0, 1, 5 and 6 are heavily constrained due to the amount of information that is available for prediction. Experiment 0 and 1, which only use air temperature, are still

able to improve predictions compared to air2stream for all model types in 7 catchments. Similarly, experiments 5 and 6 with only 6 years of training data are able to improve predictions compared to air2stream for all model types in 5 catchments.

From the results in Figure 4a,b,c it seems likely that performance is in general influenced by the combination of model, data inputs (experiment) and catchment, while the influence of different experiments and catchments is larger than the influence of model types on test RMSE. The linear regression model for test RMSE with catchment, experiment and model type as

regressors is able to explain most of the test RMSE variance with a coefficient of determination of $R^2 = 0.988$. Furthermore, it resulted in significant association of all catchments ($p < 10^{-15}$), all experiments ($p < 0.005$) and with the FNN model type ($p < 0.001$). The estimated coefficient of the FNN is -0.05, giving evidence to an prediction improvement when applying the FNN model. All other model types do not show a significant association. However, this might be due to a lack of statistical power, as the estimated coefficients of the model types (mean: -0.01, range: [-0.05, 0.02]) are generally small compared to

catchment coefficients (mean: 0.86, range: [0.69, 1.06]) and experiment coefficients (mean: -0.12, range: [-0.2, -0.04]).

Multiple experiments often result in very similar RMSE values for a single model type. Furthermore, the best performing experiment of different model types are always very close in performance. This results in a median test RMSE difference of the best experiments of different model types of 0.08 $°C$ and a median test RMSE difference of the best performing model and the second best model of another model type of 0.035 $°C$. On the other hand, the median difference between the tested

machine learning model RMSE and the air2stream RMSE is -0.39 $°C$.

### 3.3    Detailed analysis for a single catchment

To further investigate the difference in performance, the prediction results for the last year of the test data (2015) of the Inn catchment are examined. The year 2015 was chosen for comparison, since it is has an extraordinary large number of days with high water temperatures and therefore can be used to give a robust estimate of model performance. It is a strong test under

instationary conditions. The time period 1997-2014 has a median of 30 days per year with water temperatures over 11 $°C$, whereas 102 days with such high water temperature could be observed in the year 2015. Figure 5 shows the prediction results of each model (red lines) compared to the observation (blue line) and all other model predictions (grey lines) for the year 2015 and the corresponding RMSE and MAE result for that year.

The two benchmark models (LM and air2stream) show large differences between prediction and observations and show in

general a very different behaviour than all tested machine learning models. While the largest prediction errors of the tested



**Figure 4.** Boxplots of model performance comparing **a** the different machine learning models, **b** the different experiments and **c** model performance in each catchment with additional scatter plot overlay to show performance of individual combinations of catchments, models and experiments. The catchments are ordered by catchment size from smallest (left) to largest (right) with additional information of the available time series length in parenthesis below. The air2stream benchmark performance is shown as grey line for each catchment. Due to the much larger test RMSE values, LM performance is not shown to account for a better visibility.

machine learning models occur during similar time periods, large deviations can be observed over the whole year in both benchmark models.

The largest prediction errors of all machine learning models occur during warmer periods and peaks in the summer months and during periods of low water temperature in November - December. This is clearly visible in all tested models. Therefore,
differences in RMSE and MAE mainly result from their performance during these periods and consequently can be quite large even though the actual numerical difference is rather small. This can be observed when comparing the results of best performing model FNN and RNN-GRU in Figure 5. Both models produce similar prediction results for the largest part of





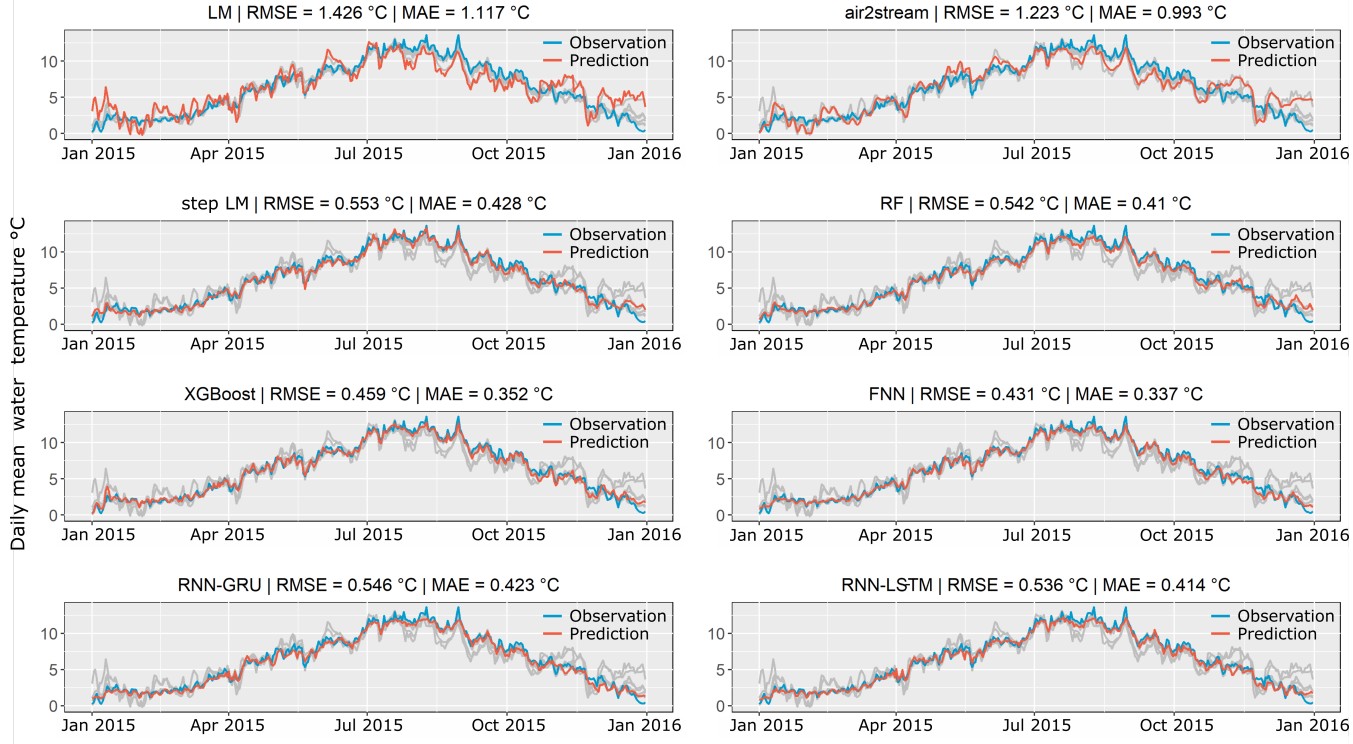

**Figure 5.** Comparison of the prediction of all tested model types for the Inn catchment for the year 2015. Data from 2015 was not used for training and validation. Prediction results for each model are shown with red lines, while the observations are shown in blue lines. The predictions of all other models are illustrated with grey lines.

the year, but the FNN model is better able to predict the peaks with high water temperatures in the summer months, which results in a RMSE and MAE difference of 0.115 and 0.086, respectively. Very small differences in RMSE and MAE as seen
between two best performing models, FNN and XGBoost, results in only very subtle differences in the predicted time series. Very similar observations can be made when analyzing the prediction results in the other catchments. The only exception can be observed in the largest catchment Donau (Figure A2), where the time series is much smoother with relatively few peaks in water temperature. This results in the RNN models being the best performing models with a large performance difference compared to all other models.

**3.4    Influence of time variables for RNNs, cross validation methods**

Removing time information in the form of fuzzy months from the training data of RNNs does not significantly change the catchment test RMSE ($p = 0.17$). However, the optimal number of time steps estimated by the hyperparameter optimization is significantly increased ($p = 0.02$). By removing time information from the inputs, the optimal timesteps are 37.78 days longer





than when using time information as additional input. This significantly increases model training time ($p = 0.034$) with a mean
difference of 132.45 minutes.

The different CV schemes applied on step LM, RF and XGBoost showed no significant difference in performance ($p = 0.91$).

### 3.5  Influence of hyperparameters on model results

The influence of different sets of hyperparameters on model performance is shown in Figure 6. This figure shows the validation
RMSE for all parameter sets which were used during hyperparameter optimization. A large difference in the range of perfor-
mance can be observed for different models. Validation RMSE means, standard deviations, minimum and maximum of all
models are shown in Table 4. The largest variability is apparent in the FNN results with a validation RMSE standard deviation
of $\sigma_{FNN} = 1.60\ °C$ and a overall RMSE range of $[0.41, 16.6]\ °C$. This is followed by XGBoost which has multiple outliers
in each catchment that increase the performance spread, resulting in $\sigma_{XGBoost} = 1.07\ °C$ and the RMSE range $[0.40, 9.15]$
$°C$. Both RNNs show a very similar performance distributions with a RMSE range of around $[0.45, 6.3]\ °C$. Compared to all
other tested models, the RF model has a much smaller spread in performance resulting from different hyperparameter sets with
$\sigma_{RF} = 0.16\ °C$ and a resulting RMSE range of $[0.45, 1.14]\ °C$.

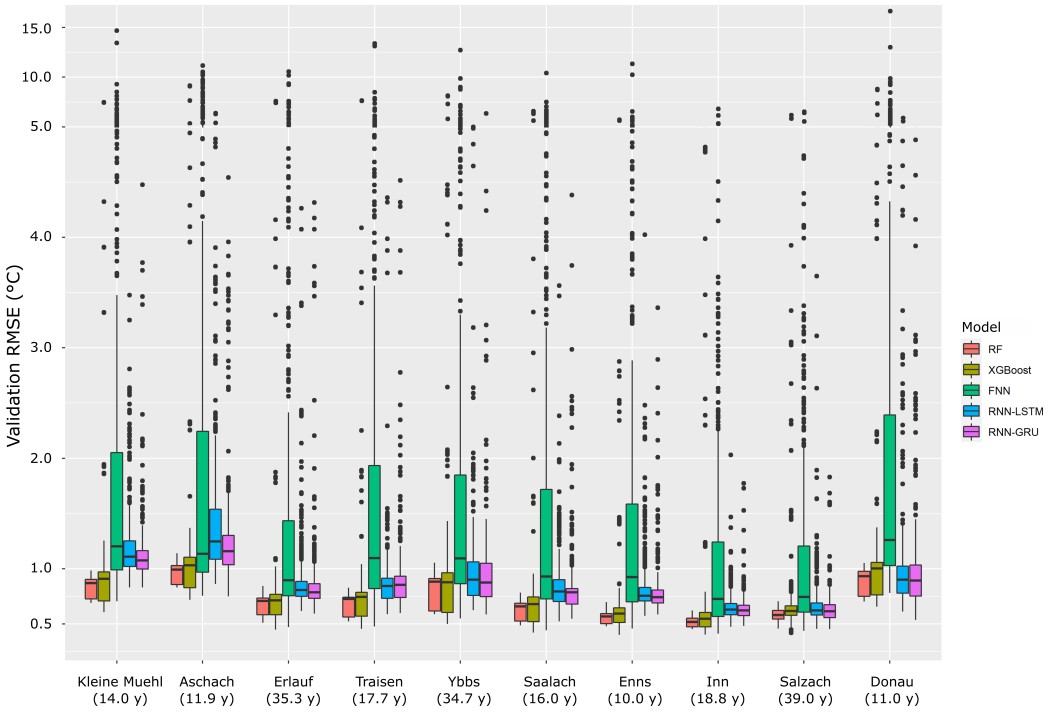

**Figure 6.** Boxplots showing the validation RMSE distribution for different hyperparameter sets for all model types, catchments and experi-
ments. The catchments are ordered by catchment size from smallest (left) to largest (right) with additional information of the available time
series length in parenthesis below.





**Table 4.** Validation RMSE means $\mu$, standard deviations $\sigma$, maxima and minima for all model types resulting from hyperparameter optimization.

| Model | Validation RMSE ($^{\circ}C$) | | | |
|---|---|---|---|---|
| | $\mu$ | $\sigma$ | min | max |
| RF | 0.70 | 0.16 | 0.45 | 1.14 |
| XGBoost | 0.95 | 1.07 | 0.40 | 9.15 |
| FNN | 1.70 | 1.60 | 0.41 | 16.6 |
| RNN-LSTM | 0.97 | 0.53 | 0.46 | 6.4 |
| RNN-GRU | 0.91 | 0.44 | 0.45 | 6.3 |

## 4 Discussion

In this study, we show the stream water temperature prediction performance of 6 machine learning models with a range of input data sets in 10 catchments and compared them to two widely used benchmark models. The results show generally a

very similar performance of the tested machine learning models with a median test RMSE difference of 0.08 $^{\circ}C$ between models. In contrast, the models had a significantly improved performance when compared to the air2stream benchmark model, with a mean test RMSE decrease of 0.42 $^{\circ}C$ (42%). Results showed that nearly all of test RMSE variance ($R^2 = 0.99$) can be explained by the catchment, the input data set and the model type. This also showed that the performance is significantly influenced by the type of input data, where more inputs generally performed better and that of all models, only the FNN

model had a significant association with lower test RMSE values. Furthermore, a wide range of performance was observed for different hyperparameter sets for the tested models, with extremely large RMSE standard deviation (1.60 $^{\circ}C$) observed in the FNN results.

Except for very few model types and experiment combinations, all tested machine learning models showed an improved performance when compared to the two benchmark models. The difference between the benchmark and the tested models was

not only visible in the resulting test RMSE and MAE values, but also clearly visible in the range and time of occurrence of large prediction errors in the predicted time series (see Figure 5). Given the range of estimated coefficients of the catchments ([0.69, 1.06]), data inputs ([-0.2, -0.04]) and model types ([-0.05, 0.02]) in the regression model for test RMSE, we can state that: given an adequate model setup and selected hyperparmeters, the influence of different data inputs and different catchments is much larger than the influence of the model types. However, there seems to be an advantage of using the FNN model, as it

was the only model that had a significant association with lower RMSE values and also the largest estimated coefficient of all model types (-0.05).

The here presented result show that FNN and XGBoost perform best in 8 of 10 catchments and are therefore a first choice for water temperature prediction tasks. For modelling large catchments with comparable size to the Donau catchment (96 000





$km^2$), where long-term dependencies seem to be more relevant, RNNs are the best choice. Both RNN architectures, GRU
and LSTM, produce very similar results in the Donau catchment, with a best test RMSE of approximately 0.52 $^{\circ}C$. This is
considerably lower than the median test RMSE of the other models (0.90 $^{\circ}C$) and the air2stream benchmark (1.10 $^{\circ}C$). The
RF model has the lowest standard deviation in resulting RMSE depending on the chosen hyperparameter (0.16 $^{\circ}C$) and thus
might be the most reasonable choice in situations with limited computational resources. More input data is generally better, but
the combination of air temperature and discharge input data already produces prediction results with a median RMSE of 0.62
$^{\circ}C$. This can be further enhanced by adding precipitation data, which decreases the median RMSE further to 0.60 $^{\circ}C$. Adding
GL data can potentially increase performance as well, as experiment 6 shows a similar performance range to experiment 3
while using only 6 years of training data. Results of experiment 2 (TP), which is most relevant for practical application as it
uses inputs that are general available for most regions and from climate models, show a median test RMSE of 0.75 $^{\circ}C$. This is
only a 19% reduction in RMSE performance compared to the experiment with the lowest median RMSE and an improvement
of 21% compared to air2stream. Thus, application with this set of widely available data inputs is able to produce prediction
performance improving the current state of the art and could be used for short term forecasts and climate change studies.

The presented machine learning approaches could considerably improve prediction results compared the current state-of-
the-art air2stream model. This stands in contrast to the findings of Zhu et al. (2019d), which assessed the performance of a
suit of machine learning models for daily stream water temperature. Zhu et al. (2019d) results showed that air2stream had an
improved performance when compared to FNNs, Gaussian process regression and decision tree models in 8 catchments using
water temperature, discharge and day-of-year as model inputs. The here presented air2stream results have a test RMSE range
of [0.74, 1.17] $^{\circ}C$, which is comparable to results of Zhu et al. (2019d) with [0.64, 1.16] $^{\circ}C$ and also to other studies applying
air2stream, e.g. Piotrowski and Napiorkowski (2018) with a range of [0.625, 1.31] $^{\circ}C$. This leads us to the conclusion that our
benchmark performance is in line with other air2stream applications and therefore provides a consistent reference, even though
air2stream was originally set up for the use of point source data and not the catchment means that we used to make results
comparable to the tested machine learning models. Consequently, our presented approaches show a significant improvement
compared to existing machine learning daily stream water temperature prediction models.

Due to the lack of physical restraints, statistical modelling approaches are often suspected to fail when extrapolating outside
their training data range (Benyahya et al., 2007). However, machine learning methods are more powerful and flexible than
previous modelling approaches and give spatial and temporal information at different scales (Reichstein et al., 2019). This
is especially important for climate change studies, where increasing air temperature might change the statistical relationships
between meteorological drivers and stream water temperature. To investigate the extrapolation performance of considered ML
methods, we selected the much warmer recent years of the time series as test period and analyzed the year with the most
frequent days of extreme temperatures in detail. All tested models where able to produce predictions with a performance close
to the training performance in the test time period and in the year with the most temperature anomalies. These results show
that it is still possible to produce robust prediction results at least for short time predictions (1-8 yrs) under a changing climate.
Successful extrapolation for short term periods suggest that also mid-to long term predictions might produce reasonable results.
However, this can only be evaluated based on future observations. It is clear that the ML approaches will fail in extrapolation,





when catchment properties change with time. In the context of high alpine, glacier dominated catchments, for example, it can be assumed that the water temperature characteristics will change, when glaciers vanish. As a consequence, the underlying processes leading the water temperature in the stream change. These changes are not reflected in the ML approaches. It would need more physically or process-based approaches. For example, air2stream would not have an advantage in this respect.

Depending on the machine learning model, our results varied considerably with the chosen hyperparameters. Especially the two best performing models, XGBoost and FNNs, show an extreme variance in performance due to the chosen hyperparameters. This leads to the conclusion that flexibility might be necessary for a well performing model, but that it is also a possible source of error or reduced model performance. These findings highlight the importance of hyperparameter optimization of machine learning models and might be a possible explanation of the fact that especially FNNs did not perform equally well in other studies. Most publications reporting findings regarding FNN performance for stream water temperature tested only a small set of FNN hyperparmeter combinations, mostly chosen by trial and error (e.g. Piotrowski et al., 2015; Rabi et al., 2015; Abba et al., 2017; Zhu et al., 2018; Temizyurek and Dadaser-Celik, 2018; Zhu et al., 2019d). Our results show the extremely large influence of hyperparmeters and therefore rendering any trial and error approach as unsuffiecient and certinly non-optimal.

RNNs are successfully applied in current rainfall-runoff modelling studies (e.g. Kratzert et al., 2018, 2019; Xiang et al., 2020; Li et al., 2020), and are thus a promising candidate for stream water prediction. However, our results show a below average performance in most catchments when compared to the other tested machine learning models. This is especially relevant, since compared to the other methods, RNNs use a range of previous timesteps (optimized hyperparameter) for prediction, which contains much more information than the 4 previous timesteps available for the other models. RNNs are the best performing models in the largest catchment indicating that RNNs are especially strong when processes with long-term dependencies have to be described. These long-term dependencies result most likely from increased concentration times, which is generally dependent on catchment size (McGlynn et al., 2004). For all other catchments in this study, the 4 day lagged variables seem to be sufficient and RNNs are not able to predict the corresponding fast changes in water temperature. Our results also show the importance of using time information as input for RNNs. RNNs are generally able to learn the corresponding information from data, since there is no significant difference in performance for the RNNs with and without time information. However, RNNs optimized with time information inputs needed a significantly lower number of timesteps for the same prediction performance, thus decreasing computation time and increasing the number of data points available for training.

This study has some limitations. Firstly, the selected catchments are all central European catchments with humid conditions. Testing these approaches on Mediterranean or more dynamic hydro-climatoligical conditions could potentially result in different importance of input variables (e.g. discharge in arid climates) and performance ranking of models. By selecting catchments with a wide range of physiographic characteristics this potential bias should be kept at a minimum. Furthermore, the performance of the air2stream benchmark is similar to the performance range of other studies, allowing for comparison. Secondly, we trained all models only for individual catchments and did not try to produce a global model that could predict water temperatures in multiple catchments, or even in a prediction of ungauged basin setting. While this is a relevant problem,





we found it necessary to have a comprehensive evaluation of different data inputs, model types and training characteristics before combining all of this in a multi-catchment water temperature prediction model.

## 5 Conclusions

Current standard methods in daily stream water prediction are able to model 10 Austrian study catchments with a mean test RMSE of $1.55°C$ (linear regression) and $0.98°C$ (air2stream). We tested 6 machine learning models with different data inputs and could produce predictions with a mean RMSE of $0.55 °C$, an improvement of 64% and 43%. Of these tested models, the FNN model using air temperature, discharge and precipitation and, if available, radiation as inputs, produce the best performing
models. With only 6 years of training data state of the art prediction models results can be achieved.

One major influence on performance are model hyperparameter. The variability in performance for different hyperparameters is much larger than for different model types or data inputs. Thus hyperparameter optimization is extremely important for a well performing model. In situations where computing resources are limited and hyperparameter optimization is not possible, the RF model seems to be a reasonable choice for application, because it has the lowest variance in prediction RMSE resulting
from the chosen hyperparameters.

RNNs with their internal states and ability to process long time series, are the best performing model type for very large catchments. This is most likely a result from increased concentration times in the catchment. Consequently, estimating concentration times of a catchment for adequately choosing a model type or relevant lags of variables should be included in future research. Applying variable importance estimation methods are also another way to further enhance the understanding of the in-
teractions between variables and model performance and could help deciding on the relevant number of variable lags. Applying these methods however, especially for neural networks, is out of scope for this study and will be part of future research.

The study catchment where chosen to have a wide range of physiographic characteristics, but are all located in Central Europe. Thus the range of characteristics is still limited and testing these model approaches in a wider range of catchments is still necessary and should also be included in future research. This will be especially important for developing multi-catchment
water temperature prediction models for regional prediction, which is an important next step and topic of current research. The presented machine learning methods, driven with observed meteorological inputs, seem to represent the system in an appropriate manner for applying them to predict river water temperature in changing conditions and may be promising for short time or real time forecasting approaches. The resulting prediction uncertainties in such systems will be mainly related to uncertainties in the meteorological forecasts. By implementing all methods into the open source R package wateRtemp,
we hope to further contribute to reproducible research and make the presented methods available and easily applicable for management issues, scientists and industries and to facilitate research on these next steps.

*Code and data availability.* The R code used to generate all results for this publication can be found in Feigl (2021b). This includes the version of the wateRtemp R package providing all machine learning methods and code that was used for producing the results of this





manuscript. A maintained and continuously updated version of the wateRtemp package can be found at github.com/MoritzFeigl/wateRtemp

or in Feigl (2021a).

We do not have the permission for further distribution of the data used in this study. All input data can, however, be acquired from the rights holders of these data sets. The water temperature and discharge data used in this study can be requested from the Central Hydrographical Bureau (HZB) at https://www.ehyd.gv.at. The rights for the meteorological data from the INCA and the SPARTACUS datasets belong to the Zentralanstalt für Meteorologie und Geodynamik (ZAMG) and can be acquired from https://www.zamg.ac.at.

**Appendix A: Model hyperparameter bounds**

**RF**: min.node.size; 2-10, mtry: 3-(number of inputs -1)

**XGBoost**: nrounds: 300-3000, eta: 0.001-0.3, max_depth: 3-12, min_child_weight: 1-10, subsample: 0.7-1, colsample_bytree: 0.7-1, gamma: 0-5

**FNN**: layers: 1-5, units: 5-200, dropout: 0-0.2, batch_size: 5-150, epochs: 100, early_stopping_patience: 5

**RNNs**: layers: 1-5, units: 5-300, dropout: 0-0.4, batch_size: 5-150, timesteps: 5-200, epochs: 100, early_stopping_patience: 5

**A1**

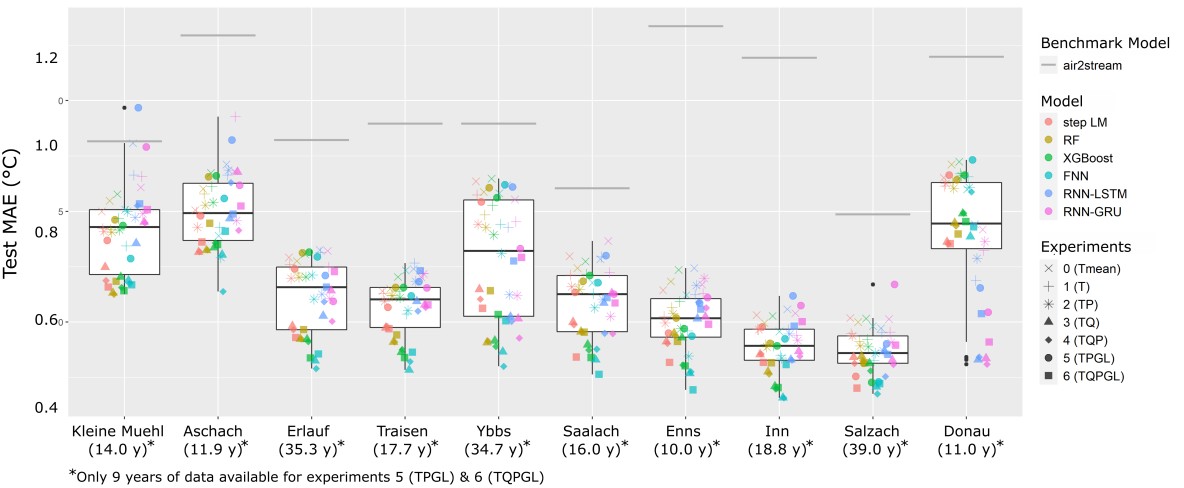

**Figure A1.** Boxplots of model performance comparing model MAE values in each catchment with additional scatter plot overlay to show performance of individual combinations of catchments, models and experiments. The catchments are ordered by catchment size from smallest (left) to largest (right) with additional information of the available time series length in parenthesis below. The air2stream benchmark performance is illustrated as grey line for each catchment.



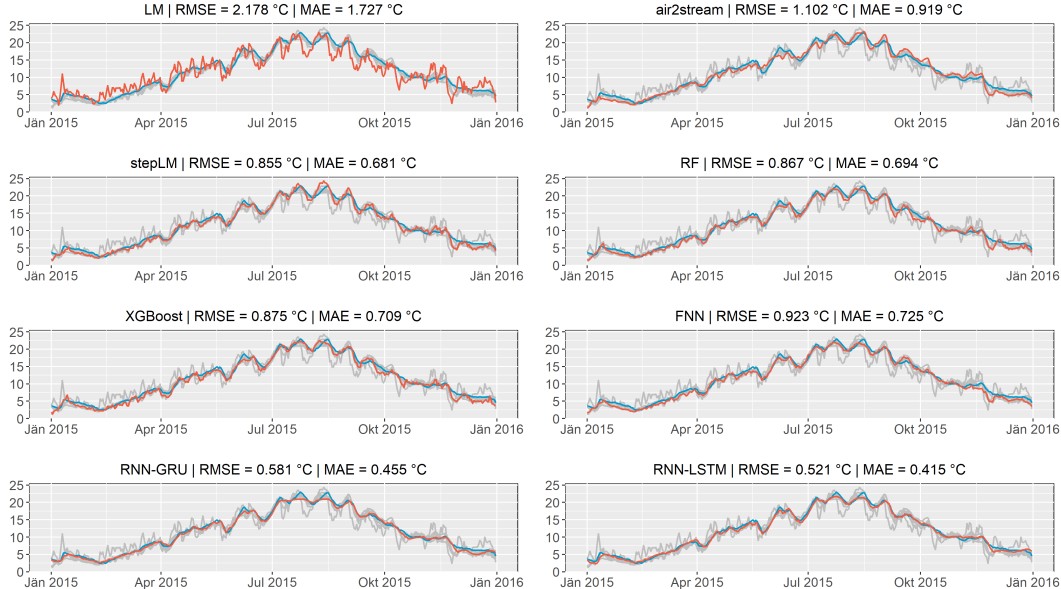

**Figure A2.** Comparison of the prediction of all tested model types for the Donau catchment for the year 2015. Prediction results for each model are shown with red lines, while the observations are shown in blue lines. The predictions of all other models are shown with grey lines.

*Author contributions.* KL, MF and MH designed the study and acquired and processed the input data. MF and KL performed all analyses and prepared the figures. MF developed the software published with this work. MH and KS contributed to the methodological framework. MF prepared the paper with contributions from KL, MH and KS.

*Competing interests.* The authors declare that they have no conflict of interest.

*Acknowledgements.* This work was partly funded by the Austrian Science Fund FWF, project number P 31213, and the Austrian Academy of Sciences (ÖAW) projects Rechout and Poco-Flood. The computational results presented have been achieved using the Vienna Scientific Cluster (VSC). We also thank Ignacio Martin Santos for providing data from the upper Danube catchment and many valuable discussions about seasonal forecast and team spirit during the Covid-19 pandemic.



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
