# Peer review of "Machine learning methods for stream water temperature prediction"

_Hydrology and Earth System Sciences, 2020_

## Referee Comment (RC1)

**Pr. Salim Heddam**
*heddamsalim@yahoo.fr.*
https://orcid.org/0000-0002-8055-8463

**General Comments**

In the present study, the authors applied and compared six machines learning (ML) models for predicting river water temperature: XGBoost, FNN, RF, two deep learning models and the step-wise linear regression. Results obtained using the proposed ML were compared to those obtained using the air2stream and the linear regression (LR). The proposed models were developed and compared using data collected at ten Austria catchments. A standard modelling approach has been adopted for this study based on linking a set of input variables to one output variable. Mean, maximal and minimal air temperatures, precipitation and global solar radiation were selected as the most relevant regressors and used as input variables for predicting water temperature. Overall the paper is very interesting, well-structured, easy to ready, and written in a scientifically sound manner. Although modelling reviver water temperature is broadly discussed in the literature, and a large amount of work has been done in recent years, admittedly with some important conclusive answers, the present work has a large potential to expand significantly our knowledge in this subject. The approach proposed in the present work owes its originality from several points, including: (*i*) new introduced ML models belonging to different categories and having different modelling strategies, (*ii*) the use of different climatic variables as input instead of what is already done in the literature (i.e., only air temperature and discharge), and (*iii*) the use of the Bayesian optimization method (BO) is innovative and the authors have developed a detail proposal. The most important finding of the present study is that the ML models were more accurate than the air2stream and the LR models, and the BO can help in improving significantly the models accuracies. In addition, the performances of the ML models varied from one catchment to another and in overall, worked equally with slightly difference. I have a number of concerns related to the paper, to my opinion needs to be clarified by the authors.

**Major's Comments**

1. The comparisons of models results with in situ measured data using only errors metrics is insufficient and does not help in providing robust conclusions regarding models accuracies, robustness and fitting capabilities. Specifically, using several kinds of goodness-of-fit indicators should be more useful: the coefficient of determination ($R^2$), the Nash-Sutcliffe efficiency (NSE), and the index of agreement *d,* are highly recommended for hydrological models evaluation (Legates and McCabe 1999; Moriasi et al. 2007; Harmel and Smith 2007; Gupta 1998, 2008; Krause et al. 2005).

2. Models structures need to be clarified. In Lines 173-175, the authors argued that including the lag of all variables for 4 previous days can help in improving models accuracies according to Webb et al. (2003). First, using only 4 previous lag should be justified, on which basis it was selected (i.e., cross-correlation analysis can be helpful for answering this question)? Second, according to Webb et al. (2003), adopting the previous lag as input variables can be useful on only hourly data scenario. Therefore, a comparison between models with and without lag data may be a good option.

**Specific Comments**

*1.* The introduction is not deeply written and in some cases need improvement. Specifically, the proposed ML reported in the literature should be presented, discussed, and the strength and weakness of each one would be more useful and effective if they are highlighted. Using lumped references do not help in understanding the mains contribution of the work.

*2.* Research gap. What are the mains contributions of the present study in comparison to what is already done? What does it add to existing literature?

*3.* Lines 47 to 50, from Austria to characteristics. To our opinion this paragraph is more suitable to be moved to section 2.1.

*4.* Line 79: ''*To the author's knowledge, RF has not been applied for river water temperature prediction yet*''. This statement is incorrect. The RF was recently reported as a powerful tool for predicting river water temperature (Heddam et al. 2020).

*5.* Models comparison using cross-station scenarios can help in providing more conclusions, and a clear idea about models capabilities outside of their own catchment area: models calibration using data from on station and validated for other stations (i.e., see Zhu and Heddam 2019).

**References**

Gupta, H.V., Sorooshian, S., Yapo, P.O., (1998). Toward improved calibration of hydrological models: multiple and non-commensurable measures of information. Water Resour. Res. 34, 751-763.

Gupta, H.V., Wagener, T., Liu, Y., (2008). Reconciling theory with observations: elements of a diagnostic approach to model evaluation. Hydrol. Proc. 22, 3802-3813.

Harmel, R.D., & Smith, P.K. (2007). Consideration of measurement uncertainty in the evaluation of goodness-of-fit in hydrologic and water quality modeling. *Journal of Hydrology*, *337*(3-4), 326-336.

Heddam, S., Ptak, M., & Zhu, S. (2020). Modelling of daily lake surface water temperature from air temperature: Extremely randomized trees (ERT) versus Air2Water, MARS, M5Tree, RF and MLPNN. *Journal of Hydrology*, *588*, 125130.

Krause, P., Boyle, D., Bse, F., (2005). Comparison of different efficiency criteria for hydrological model assessment. Adv. Geosci. 5, 89-97.

Legates, D.R., & McCabe Jr, G.J. (1999). Evaluating the use of "goodness-of-fit" measures in hydrologic and hydroclimatic model validation. *Water resources research*, *35*(1), 233-241.

Moriasi, D.N., Arnold, J.G., Van Liew, M.W., Bingner, R.L., Harmel, R.D., & Veith, T.L. (2007). Model evaluation guidelines for systematic quantification of accuracy in watershed simulations. *Transactions of the ASABE*, *50*(3), 885-900.

Webb, B.W., Clack, P.D., Walling, D.E. (2003). Water-air temperature relationships in a Devon river system and the role of flow, Hydrological Processes, 17, 3069-3084.

Zhu, S., & Heddam, S. (2019). Modelling of maximum daily water temperature for streams: Optimally pruned extreme learning machine (OPELM) versus radial basis function neural networks (RBFNN). *Environmental Processes*, *6*(3), 789-804.

---

## Author Response (AR1)

**Review Answers**

**Reviewer 1**

Dear Salim Heddam,

We thank you for your encouraging and positive feedback and sincerely thank you for your insightful comments and suggestions. Please find our answers to your comments below. We highly acknowledge and thank you for your review, comments and time and have stated this in the acknowledgements section of the manuscript.

**Review:**

The comparisons of models results with in situ measured data using only errors metrics is insufficient and does not help in providing robust conclusions regarding models accuracies, robustness and fitting capabilities. Specifically, using several kinds of goodness-of-fit indicators should be more useful: the coefficient of determination ($R^2$), the Nash-Sutcliffe efficiency (NSE), and the index of agreement d, are highly recommended for hydrological models evaluation (Legates and McCabe 1999; Moriasi et al. 2007; Harmel and Smith 2007; Gupta 1998, 2008; Krause et al. 2005).

**Answer:**

We agree that by choosing a variety of metrics, a more concise picture of model performance can be shown. However, we noticed that some metrics are not sensitive enough to compare the results of this study. The NSE values of the presented models were all >0.9 and usually around 0.98. We noticed that while we could still see differences in model performance in RMSE and MAE, there were no or hardly any differences in the first three decimals of the NSE. Similar observations were made for the coefficient of determination and the index of agreement. Therefore, we think that adding these metrics would decrease readability while not adding a lot of information, but we added them to the appendix table A1 to allow for future comparisons.

Here is an overview of all metrics for the best ML models and the two benchmark models per catchment:

| Catchment | Model | Best ML model results | | | | | LM | | | | | Air2stream | | | | |
|---|---|---|---|---|---|---|---|---|---|---|---|---|---|---|---|---|
| | | RMSE | MAE | NSE | d | R2 | RMSE | MAE | NSE | d | R2 | RMSE | MAE | NSE | d | R2 |
| Kleine Mühl | XGBoost | 0.740 | 0.578 | 0.982 | 0.995 | 0.983 | 1.744 | 1.377 | 0.899 | 0.971 | 0.903 | 0.908 | 0.715 | 0.973 | 0.993 | 0.974 |
| Aschach | XGBoost | 0.815 | 0.675 | 0.983 | 0.996 | 0.983 | 1.777 | 1.408 | 0.920 | 0.978 | 0.924 | 1.147 | 0.898 | 0.969 | 0.992 | 0.970 |
| Erlauf | XGBoost | 0.530 | 0.419 | 0.985 | 0.996 | 0.986 | 1.345 | 1.057 | 0.884 | 0.968 | 0.900 | 0.911 | 0.717 | 0.959 | 0.989 | 0.960 |
| Traisen | FNN | 0.526 | 0.392 | 0.985 | 0.996 | 0.985 | 1.254 | 0.970 | 0.912 | 0.977 | 0.915 | 0.948 | 0.757 | 0.951 | 0.988 | 0.955 |
| Ybbs | RF | 0.576 | 0.454 | 0.989 | 0.997 | 0.989 | 1.787 | 1.415 | 0.889 | 0.971 | 0.890 | 0.948 | 0.744 | 0.968 | 0.992 | 0.969 |
| Saalach | XGBoost | 0.527 | 0.420 | 0.977 | 0.994 | 0.979 | 1.297 | 1.062 | 0.864 | 0.961 | 0.883 | 0.802 | 0.633 | 0.951 | 0.988 | 0.955 |
| Enns | FNN | 0.454 | 0.347 | 0.984 | 0.996 | 0.985 | 1.425 | 1.166 | 0.834 | 0.951 | 0.840 | 0.835 | 0.671 | 0.946 | 0.986 | 0.952 |
| Inn | FNN | 0.422 | 0.329 | 0.984 | 0.996 | 0.984 | 1.376 | 1.098 | 0.829 | 0.949 | 0.830 | 1.170 | 0.952 | 0.882 | 0.968 | 0.882 |
| Salzach | FNN | 0.430 | 0.338 | 0.986 | 0.996 | 0.986 | 1.327 | 1.077 | 0.862 | 0.961 | 0.864 | 0.743 | 0.589 | 0.957 | 0.989 | 0.963 |
| Donau | RNN-LSTM | 0.521 | 0.415 | 0.986 | 0.996 | 0.989 | 2.145 | 1.721 | 0.842 | 0.955 | 0.843 | 1.099 | 0.911 | 0.961 | 0.990 | 0.968 |

We used the index of agreement by Willmott, 1981.

References:

Willmott, C. J. (1981). On the validation of models. *Physical Geography*, 2(2), 184–194. https://doi.org/10.1080/02723646.1981.10642213

**Review:**

Models structures need to be clarified. In Lines 173-175, the authors argued that including the lag of all variables for 4 previous days can help in improving models accuracies according to Webb et al. (2003). First, using only 4 previous lag should be justified, on which basis it was selected (i.e., cross-correlation analysis can be helpful for answering this question)? Second, according to Webb et al. (2003), adopting the previous lag as input variables can be useful on

only hourly data scenario. Therefore, a comparison between models with and without lag data may be a good option.

**Answer:**

This is indeed an important point and we agree that we should explain our reasoning. Our decision on the number of lags was based on the results of an explorative data analysis of daily data we carried out before starting to work on the model structures. This included three major parts that guided our decision:

1. Assessing partial autocorrelation plots of water temperatures: They showed a significant importance usually up to the 4th lag, as is illustrated in the following plot showing the partial autocorrelation of stream water temperatures of the Ybbs catchment.

[Figure]

2. Assessing variable significance in a linear regression model using multiple lags: This also pointed to the fact that 4 lags are an adequate range for the given basins.
3. Assessing variable importance in a simple Random Forest Model, which also pointed to the fact that 4 lags are a reasonable choice, as is visible from the following plot showing the RF variable importance for predicting water temperatures in the Ybbs catchment.

[Figure]

Our initial analysis pointed to the fact that lags are important and that 4 is a good choice for our set of basins, thus resulted in our decision for this study. Nevertheless, this is a relevant initial decision. Our results let us assume that this might be quite different for different basins and maybe a more dynamic approach might be valuable. This could for example be including the choice of lag time depending on the mean concentration time of a catchment, or the catchment size and should be explored in future studies.

We added a short summary of these initial analyses after Line 173 to explain our choice of numbers of lags:
"The lag period of 4 days was chosen based on an initial data analysis that included (i) assessing partial autocorrelation plots of water temperatures, (ii) testing for significance of lags in linear regression models, and (iii) checking variable importance of lags in a random forest model."

Regarding the study of Webb et al. (2003): While they only showed the importance of lagged variables for hourly data, they also summarized findings regarding lagged air temperature for daily data (Grant, 1977; Jeppesen & Iversen, 1987; Stefan & Preud'homme, 1993). These previous findings, together with our initial analyses (especially the RF variable importance), made it clear that daily lags are relevant inputs and necessary for a high model performance.

References:
Grant, P. J. (1977). Water temperatures of the Ngaruroro river at three stations. *Journal of Hydrology*, *16*(2), 148–157. https://www.jstor.org/stable/43944413?seq=1
Jeppesen, E., & Iversen, T. M. (1987). Two Simple Models for Estimating Daily Mean Water Temperatures and Diel Variations in a Danish Low Gradient Stream. *Oikos*, *49*(2), 149. https://doi.org/10.2307/3566020
Stefan, H. G., & Preud'homme, E. B. (1993). Stream temperature estimation from air temperature. *JAWRA Journal of the American Water Resources Association*, *29*(1), 27–45. https://doi.org/10.1111/j.1752-1688.1993.tb01502.x
Willmott, C. J. (1981). On the validation of models. *Physical Geography*, *2*(2), 184–194. https://doi.org/10.1080/02723646.1981.10642213

**Review:**
The introduction is not deeply written and in some cases need improvement. Specifically, the proposed ML reported in the literature should be presented, discussed, and the strength and weakness of each one would be more useful and effective if they are highlighted. Using lumped references do not help in understanding the mains contribution of the work.

**Answer:**
Thank you for pointing this out, comparing the different models instead of only giving an overview of past applications will make this section more informative. We added high-level descriptions and general advantages/disadvantages of all models in the revised Introduction. Furthermore, we made a more detailed overview of past FNN applications.

**Review:**
*Research gap. What are the mains contributions of the present study in comparison to what is already done? What does it add to existing literature?*

**Answer:**

We agree, this might not be stated clearly enough in the manuscript yet. We think that the summary of the important contributions of this study in your general comments are very much on point. We include it in the last paragraph of the introduction, by changing the sentence "This includes novel approaches …" in line 125 to:

"The present work's originality includes (i) application of a range of ML models for water temperature prediction, (ii) the use of different climatic variables and combinations of these as model inputs, and (iii) the use of Bayesian optimization to objectively estimate hyperparameters of the applied ML models."

**Review:**

Lines 47 to 50, from Austria to characteristics. To our opinion this paragraph is more suitable to be moved to section 2.1.

**Answer:**

Agreed, we changed it according to your suggestion. This paragraph is now at the beginning of section 2.1.

**Review:**

Line 79: ''*To the author's knowledge, RF has not been applied for river water temperature prediction yet''.* This statement is incorrect. The RF was recently reported as a powerful tool for predicting river water temperature (Heddam et al. 2020).

**Answer:**

Thank you for pointing this out. This is indeed an interesting and relevant publication. We changed the sentence in line 79 to the following:
"Up to date, only one previous study by Heddam et al. (2020) applied RF for predicting lake surface temperatures."

**Review:**

Models comparison using cross-station scenarios can help in providing more conclusions, and a clear idea about models capabilities outside of their own catchment area: models calibration using data from on station and validated for other stations (i.e., see Zhu and Heddam 2019).

**Answer:**

We do agree that the application outside the initial trained catchment is an important type of application. However, we found it necessary to only focus on the model prediction capabilities in single catchments, to derive the general applicability of different model types and data inputs. These should be used as a foundation to derive transferable models and modelling approaches. The transferability of these models in this study cannot be adequately tested, as there is no information provided to the tested models to conduct this transfer. In our opinion, this transfer would need additional basin characteristics as inputs and consequently a larger number of basins for training and testing (multi-basin training). We certainly do see this as an important next step, but would refrain of applying the single basin trained models for this task. We thank the reviewer for this thought and added this topic in the conclusions regarding future research fields in Line 615:
"The development of regional models would also need to include comparison of cross-station scenarios and other tests for model transferability in time and space."

**Reviewer 2**

Dear Adrien Michel,

we thank you for your time and effort, bringing up these thoughtful questions and sincerely thank you for your insightful comments and suggestions on the manuscript. To address them more easily we split them into several parts. Please find our answers to your comments below. We highly acknowledge and thank you for your review, comments and time and have stated this in the acknowledgements section of the manuscript.

**Review:**

Impact of snow/glacier cover and catchment size

The authors discuss the importance of snow/glacier cover in the perspective of climate change (CC) application (I discuss CC below). However, this is not discussed for the application done in the paper. First, I suggest to add in Table 1 the mean catchment elevation and percentage of glacier cover in the catchments in order to allow a quick overview of the contribution of glacier and snow melt we can expect on each catchment.

**Answer:**

These points are highly interesting and will be addressed in more detail in a subsequent publication, which is currently in preparation. To answer your questions here, we computed the percentage of glacier and perpetual snow cover from the CORINE Land Cover data 2012 and the mean catchment elevation from the EU-DEM v1.1 digital elevation model with 25x25 m resolution, which we included in Table 1 of the revised manuscript. The following table shows the corresponding values for all study catchments and also the glacier and perpetual snow cover in km²:

| Catchment | Glaciers and perpetual snow (km²) | Glaciers and perpetual snow (% of total catchment area) | Mean catchment elevation (m NAP) |
|---|---|---|---|
| Kleine Mühl | 0.0 | 0 | 602 |
| Aschach | 0.0 | 0 | 435 |
| Erlauf | 0.0 | 0 | 661 |
| Traisen | 0.0 | 0 | 697 |
| Ybbs | 0.0 | 0 | 691 |
| Saalach | 0.0 | 0 | 1196 |
| Enns | 0.1 | 0.003 | 1419 |
| Inn | 60.0 | 2.8 | 2244 |
| Salzach | 60.3 | 1.4 | 1475 |
| Donau | 339.0 | 0.4 | 827 |

**Review:**

I would expect the TQ experiments to perform significantly better than the TP ones in catchment where snow plays an important role. Indeed, snow melt dynamic is captured in Q while I doubt TP experiments will be able to get it. This is difficult to see from Fig 4c and A1, so I would suggest to add further information about TP vs TQ comparison in high elevation catchment. Especially since the authors mention TP performance as an argument for usage in CC impact studies (lines 532 to 537).

**Answer:**
We believe that these are important questions and we again want to refer to our subsequent publication for a detailed answer. To give an overview here, we analyzed the differences in TP and TQ performances depending on elevation and glacier cover. TQ generally shows a significantly lower test RMSE. The difference between TQ and TP performance at the same time does not show any changing patterns when evaluating high elevation catchments in our data. This is visible in the following figure, which contains test RMSE values for both experiments 2(TP) and 3(TQ) for all catchments. The catchments are sorted by elevation and glacier cover fraction, from left (low elevation, no glacier) to right (high elevation, glacier).

[Figure]

While we do not observe the relationship you expected, it is also important to keep in mind that our study catchments are mainly larger catchments and limited in their number (n = 10), They were also intentionally chosen to be quite different from each other. Therefore, it might be difficult to test this hypothesis, since we lack the necessary number of small high elevation catchments.

As you expected, there is a significant relationship between mean catchment elevation, glacier fraction and test RMSE. We tested this with a linear model using mean catchment elevation, glacier fraction in % of the total catchment area, total catchment area and the experiments as independent variables and test RMSE as dependent variable. This is illustrated in the following figure which shows test RMSE values and mean catchment elevations for all experiments, with additional information of the glacier area.

[Figure]

We added a sentence regarding the relationship between test RMSE and mean elevation in the results section 3.2 as described in the answer to the third comment.

**Review:**

Catchment's size seems to have a clear influence on the results. Indeed, if we neglect the Danube catchment, Figures 4c and A1 show a reduction of RMSE and MAE with increasing catchment's size. This is not surprising since I would expect local scale effects, harder to capture in models, to be smoothed out when increasing catchments size, leading to an increase of the model performance. It could be interesting to replace catchment with catchment size in the linear regression for test RMSE, or use both, in order to really asses it (in any case this would mean a regression with both discrete and continuous variables). This size effect is currently not discussed (except for the Danube) while I think it should definitely be mentioned.

**Answer:**

We agree, catchment size seems to have an effect on model performance. When substituting catchment with catchment area in the linear regression model for test RMSE, we find a significant (p-value $= 3.91 \times 10^{-4}$) influence. This is also visible when plotting test RMSE of all experiments together with the logarithm of the catchment area:

[Figure]

Overall, we agree that due to aggregation, local small scale effects on stream water temperature are smoothed out, making it easier to perform predictions based on catchment means. The fact that the Danube shows a reduced performance instead of an increased one for all models except RNNs, leads us to the conclusion that while larger catchments are easier to model, they need additional lagged information due to longer-term dependencies.

We added the findings regarding catchment area, elevation and glacier cover in the results section 3.2 after line 460:
"The relationship between mean catchment elevation, glacier fraction and test RMSE was analyzed with a linear model using mean catchment elevation, glacier fraction in % of the total catchment area, total catchment area and the experiments as independent variables and test RMSE as dependent variable. This resulted in a significant association of elevation (p-value $< 2 \times 10^{-16}$) with lower RMSE values and catchment area (p-value $= 3.91 \times 10^{-4}$) and a significant association of glacier cover (p-value $= 9.79 \times 10^{-5}$) with higher RMSE values. Applying the same model without using the data of the largest catchment, the Danube, resulted in a significant (p-value $= 2.12 \times 10^{-11}$) association between catchment area and lower RMSE values, while the direction of the other associations stayed the same."

Furthermore, we added a statement in the discussion at the end of line 562:
"The current results suggest a strong influence of catchment properties on general ML model performance. While association of performance with elevation, glacier cover and catchment area were apparent, we could not come to a conclusion as even the direction of the relationship for one variable changed when removing one catchment from the analysis. We believe that there are a number of factors influencing these associations and more in depth investigations on a larger number of basins are needed to further understand the relationships between ML model performances and catchment properties and their implications. "

**Review:**
As mentioned above, I have no expertise in the ML domain beyond basic knowledge. As a novice, I found the Section 2.4 quite technical (especially Section 2.4.5). Having in mind the target audience of HESS, I would suggest to keep in the main text an overview of the different ML models used, and to move the most technical parts in Supplementary Material along with the details requested below.

**Answer:**

Thank you for this important comment, we agree that section 2.4 is a bit technical. According to your suggestion, we moved the RNN cells description to the Appendix. Furthermore, we added high-level descriptions and general advantages/disadvantages of all model in the revised Introduction. This should allow readers to gain an overview of the model properties, while keeping a somewhat detailed description of the models in the methods section.

**Review:**

This would allow the authors to present more details about the reproducibility of the study which are not yet presented. Indeed, Appendix A only shows the hyperparameters bounds used for the Bayesian optimization. The final set of hyperparameters should be provided (along with the parameters of the two benchmark models). It is not completely clear for me if the Bayesian optimization is done in general or per catchment. This should be stated. Also, is the optimization run for each separate experiment, or only once? And in this case for which experiments? If the Bayesian optimization is done per catchment separately, is not it a risk of overfitting? In summary, some details and clarity are missing about how the Bayesian optimization is done (which catchments, experiments and time periods), and how the models' training is done. Note that the calibration procedure of air2stream should also be presented.

Thank you for raising this point. We agree that providing the estimated hyperparameter sets in the supplementary material would be helpful and we included these in the revised version. They are referred to in an additional sentence in line 501:
"Tables with all optimized hyperparameters are provided in the supplementary material."

We would like to note that the code we used and provided for reproducibility purposes included fixed seeds in the random number generation to make our results reproduceable. Therefore, by running the provided code, anyone should be able to reproduce our results on any computer.
Since we are only investigating single basin models, Bayesian hyperparameter optimization was applied for each combination of catchment, model, and experiment. Should there be any overfitting, it would result in a lower performance in the test time period, which was never used for any model selection or training. Using cross-validation and training/validation splits during training should prevent potential overfitting.
To make the whole optimization procedure and overall training setup more comprehensible, we added a paragraph and a table at the end of section 2.7(Experimental setup). While the paragraph includes a summary of the whole modelling/optimization procedure (including the air2stream optimization), the table contain the following information: catchment name, training/validation time period, test time period, cross-validation yes/no, number of hyperparameters, number of iterations of hyperparameter tuning.

New paragraph & table:
"Table 3 gives an overview of all time periods and the hyperparameter optimization details. All models are trained using the training/validation period data and either applied CV or a training/validation split. Models with hyperparameters are trained multiple times during hyperparameter optimization. The fully trained models are then applied in the test time period to produce comparable out-of-sample results. The 8 air2stream hyperparameters are optimized using the Particle Swarm Optimization with 500 iterations, 500 particles, cognitive and social learning factors set to 2 and inertia max and min set to 0.9 and 0.4. All models were run on the Vienna Scientific Cluster, where each run had access to 2 Intel Xeon E5-2650v2, 2.6 GHz, 8 cores CPUs and 65 GB RAM."

**Table 3.** Overview of the different modelling time periods and hyperparameter optimization details, including information about cross-validation (CV), the number of hyperparameters (Hypeparameters) and the number of iterations of the bayesian hyperparameter optimization (Iterations).

| Catchment | training/validation period | test period | Model | CV | Hyperparameters | Iterations |
|---|---|---|---|---|---|---|
| Kleine Mühl | 2002-2012 | 2013-2015 | LM | no | 0 | 0 |
| Aschach | 2004-2012 | 2013-2015 | air2stream | no | 8 | 500 |
| Erlauf | 1980-2007 | 2008-2015 | step-LM | yes | 0 | 0 |
| Traisen | 1998-2011 | 2012-2015 | RF | yes | 2 | 60 |
| Ybbs | 1981-2007 | 2008-2015 | XGBoost | yes | 6 | 60 |
| Saalach | 2000-2011 | 2012-2015 | FNN | no | 4 | 60 |
| Enns | 2006-2013 | 2014-2015 | RNN-GRU | no | 5 | 60 |
| Inn | 1997-2011 | 2012-2015 | RNN-LSTM | no | 5 | 60 |
| Salzach | 1977-2007 | 2008-2015 | | | | |
| Danube | 2005-2012 | 2013-2015 | | | | |

**Review:**
The computational cost seems to be a major concern using ML models and is mentioned multiple times throughout the paper. It would be interesting to have indications about the hardware used and the total time needed for the Bayesian optimization, the learning phase, and the running phase along with the time needed to calibrate and run the two benchmark models. This would help the reader to apprehend the computational implication of using ML models.

**Answer:**
We absolutely agree with this remark. We added an additional paragraph and table with median and inter quartile range of the run time for each ML model in the results section 3.2.: "The run times for all applied ML models are summarized in table 5 FNN and RF have the lowest median run times with comparatively narrow inter quartile ranges (IQR), so that most models take between 30 minutes and one hour to train. XGBoost has a median run time of around 3 hours (172.9 min) and also a comparatively low IQR with a span of 50 min. step LM and both RNNs need much longer to train with median run times around 700 minutes. They also have a much larger variability in the needed run time, especially the step LM model with an IQR of more than 1500 minutes. In contrast, the run time of the LM model is neglectable small (< 1 sec) and air2stream is also considerably faster with run times of < 2 min in all catchments."

**Table 5.** Run times of all applied ML models given as the median and inter-quartile ranges (IQR) of run times in minutes.

| Model | Run times (min) | |
|---|---|---|
| | median | IQR |
| step LM | 698.9 | 158.8 - 1733.8 |
| RF | 54.3 | 44.3 - 74.6 |
| XGBoost | 172.9 | 153.6 - 204.0 |
| FNN | 30.8 | 28.5 - 41.5 |
| RNN-LSTM | 748.6 | 520.9 - 1111.6 |
| RNN-GRU | 767.8 | 583.9 - 1171.1 |

We added details about the hardware used in section 2.7 (Experimental setup):
"All models were run on the Vienna Scientific Cluster, where each run had access to 2 Intel Xeon E5-2650v2, 2.6 GHz, 8 cores CPUs and 65 GB RAM. "

**Review:**

Finally, the training is done on really different time period lengths. The results seem to suggest that there is no correlation between length of the training period and RMSE. This difference in period length is not really discussed in the paper. In general, having similar training periods would be beneficial to really compare the models' performances across catchments. Indeed, with the data provided, we do not know if the differences observed across catchments are due to catchments' characteristics or to the training time period length. I imagine that the heavy computational time forbids to re-run all catchments using similar datas. However, a re-run on a single catchment with > 30 yrs of data, but using only 10 years as for the Enns, could be interesting to assess the impact of the length of the training time series on the results. Note that this question of length of the time series available for training is an important point for application perspective. Indeed, water temperature measurements network are usually quite recent (few decades), compared to time series available for discharge.

**Answer:**

From the current results we assume not much difference in performance between a model using 30 years and a model using 10 years of data. Our results show that also short time series (6 years of training in all experiment 5 & 6 models) can produce state of the art results. It is nevertheless true that we do not know the data length threshold at which the model performance is significantly reduced. However, we believe that this threshold is highly dependent on the catchment and the main processes that are present in the catchment. We share your interest in these points and thank you for your suggestions. However, at this point we would like to kindly point out to the follow-up study we are currently working on.

As you mentioned, there is no apparent correlation between RMSE and the length of the training period in our results. Based on these results we conclude that the influence of the catchment is most likely larger than the influence of the time series length. However, we believe that, since the number of investigated catchments is limited, effects such as the relationship of time series length and model performance can not be estimated with certainty. There may be a strong correlation between time series length and RMSE in certain types of catchments (e.g. small alpine catchments), but finding these relationships is not possible with the given study setup. Thus, we would like to refrain from making assumptions we cannot yet proof.

**Review:**

The Section 3.3 focuses on the catchment obtaining the second lowest RMSE. It would be interesting to perform a similar analysis for a catchment obtaining higher RMSE. For example, analyzing the Kleine Muehl, where the median of ML RMSE is close to air2stream, would be interesting. In any case, some plots of the whole time series for all catchments should be presented in Supplementary in order to allow the reader to look at the real outputs and not only have access to RMSE boxplots (I mention here that I do particularly appreciate Figure 5 where all outputs are always shown in grey in the background).

**Answer:**

As suggested, we added the time series plots of all other catchments in the supplementary material and refer to them at the end of section 3.3:

"The corresponding figures of all catchments except the Inn and Danube are provided in the supplementary material."

We agree that there are additional aspects of these results that would also be interesting to further investigate and discuss, but as the manuscript was already quite extensive, we only chose one catchment. We hope that adding the remaining catchments in the supplementary material is a reasonable compromise. Please find the time series of the Kleine Mühl catchment below. As you can see, while air2stream is indeed performing much better in the Kleine Mühl compared to the Inn catchment, but the overall characteristics of the models are quite similar.

[Figure]

**Review:**

Two main applications are mentioned: short term predictions (especially high summer water temperature peak) and climate change (CC). For the first application, further discussion could be added in Section 3.3. Indeed, In Figures 5, we do see that many short timed high temperature events are not captured during the summer, which would be problematic for predictive application. Metrics do exist to assess the quality of models to capture such events (see e.g. the two-alternative forced choice score used in Greissinger et al. (2019)). While this paper of course does not pretend to deliver operational models – I rather see the paper as a demonstration of current capabilities of ML models for water temperature predictions along with pros and cons of different models – a discussion of the performance of ML models regarding short term high temperature event would be a great addition.

**Answer:**

This is indeed an important point and we agree that our description of the possible application was not precise enough. In general, we believe it is very important to differ between simulation and forecast models (Beven & Young, 2013). While simulation models aim to predict a certain variable (e.g. water temperature) by using inputs that reflect our process understanding, forecast models use all available previous information, including previous observations of this variable. While simulation models can show important connections between input variables and the output variable and can be used to learn more about a system, forecast models will most likely always provide a better prediction performance. As you mention, we do not aim to provide the best possible operational (i.e. forecasting) model, but

investigate model types, input data and training procedures for water temperature (simulation) modelling. Based on these results, developing an operational forecasting model would mainly consist of choosing the model type and data depending on the aims and availability and include water temperature from previous time steps as an additional input.

We adapted our statement on short term forecast in the discussion (line 536) and used "… as basis for short term forecast .." instead of "…for short term forecast .."

**Reference:**

Beven, K. and Young, P. (2013) 'A guide to good practice in modeling semantics for authors and referees', *Water Resources Research*. Blackwell Publishing Ltd, 49(8), pp. 5092–5098. doi: 10.1002/wrcr.20393.

**Review:**

The author choose an interesting approach to assess the ability of the model to cope with non-stationarity in time series by using the CC signal present in the measurements timeseries. They show that ML models still obtain good performance in the warm year 2015. This could be enhanced by showing the whole time series in order to see if the error grows with time (and to better compare with benchmark models). Using one catchment with >30 yrs time series and train it only with the first 10 years (as suggested above to assess the effect of training time period length) could also be interesting in this regard. Indeed, the temperature increase between the 80's and 2015 will be more important than the one in the time series of the Inn catchment used in the paper to do this validation. Note that the years 2003, which shows an important water temperature anomaly, is also an interesting benchmark year.

**Answer:**

We actually only noticed that the ML models were able to cope with signals in the observations, which were outside the range of the training data, after all model simulations were performed. We agree that assessing the performance of a model trained with data from the 80's with data from 2015 would be a much harder and more comprehensive test. We hope for your understanding, that this test cannot be added to the already extensive manuscript. There are three important reasons: Only three catchments have time series that are long enough (> 30 years) leaving us with a very limited sample size. Secondly, data collection technology was updated at some point in the 90's for all catchments. While this is not a large source of error for the current model setup, where enough data points in later years are available, it can be assumed that this will most likely reduce performance. Consequently, additional analysis of the underlying observations is necessary to better estimate the origin of potential modelling errors. Thirdly, the manuscript focuses on the comparison of several ML approaches and is already quite extensive. Adding the "CC-test" would therefore be beyond the scope for the current manuscript.

**Review:**

While these tests are important and showing that models are able to correctly predict water temperature when out of the training range is a really good point for ML models, the increase of water temperature expected with CC is far above the range tested here. As a consequence, I do not think that it is really possible to assess ML models' ability to correctly predict CC impacts on water temperature with historical data. A comparison with physically based models could be an approach, but is beyond the scope of this work. Consequently, I would suggest to revise the lines 535 and 536 in the discussion.

**Answer:**

We agree that this statement is too general and is not supported by our findings. We changed it to: "Thus, application of this set of widely available data inputs is able to produce prediction performance improving the current state of the art and could be used as basis for short term forecasts and assessing near future predictions (5-10 years) under climate change. The ability of ML approaches to simulate processes and signals from a system under prolonged climate change is important and topic of future research."

**Review:**
Figure 1: Please specify where you obtained the catchments delineation. Also, the figure mention "Danube catchment" while Donau is used elsewhere in the paper. Maybe the English name Danube should be used, it would be more accessible to international readers.
**Answer:**
We added the delineation sources to the description of Figure 1: Bayrisches Landesamt für Umwelt; HAO: Hydrological Atlas of Austria (digHAO), 3. Delivery, Federal Ministry of Agriculture, Regions and Tourism, Vienna, Austria, 2007 and changed "Donau" to "Danube" in the manuscript text and all figures as you suggested.

**Review:**
Table 1: In addition to what is already mentioned above, add calibration and testing periods to the table.
**Answer:**
We included them in the additional table in section 2.7 (Experimental setup) that we added in regards to your major comment.

**Review:**
Line 48: date -> data
**Answer:**
The suggested change was included in the revised manuscript.

**Review:**
Lines 67-69: "Another main concern is that parametric statistical models showed higher prediction performances on weekly, monthly or seasonal time scales in the past (Caissie, 2006) leading to a loss of temporal variation (DeWeber and Wagner, 2014)."
Higher prediction performances compared to what (models and/or temporal resolution)?
**Answer:**
In this section, we give an overview about strengths of different modelling types, which were used for river water temperature modelling in the past. With this sentence we aimed to point out that parametric statistical models achieved higher performances on coarser time scales than on finer temporal resolution – especially when considering air temperature as solely surrogate. However, after further examination we came to the conclusion that it is indeed a bit misplaced and does not add any important information to the paragraph and thus decided to remove it from the revised manuscript.

**Review:**
Line 177: moths -> months
**Answer:**
The suggested change was included in the revised manuscript.

**Review:**
Line 255: Misplaced parenthesis and inversed sum bounds in eq (5)

**Answer:**
The suggested change was included in the revised manuscript.

**Review:**
Section 2.4.5: There is an overall inconsistency between parts of the text and equation in the usage of bold font for vector and matrix terms. E.g. in the paragraph at lines 308-310 they are in bold, while in the following paragraph they are in italic. I would suggest the usage of bold fonts everywhere.
**Answer:**
Thank you for this observation and comment, we adapted the formulas as you suggested.

**Review:**
Line 329: Hyperparameters meaning is never really defined in the paper
**Answer:**
We added an additional sentence in line 117: "The term "hyperparameters" refers to any model parameter that is chosen before training the model (e.g. Neural network structure)".

**Review:**
Line 356: The difference between validation and testing periods is not really clear. I understand the validation is used to choose the best models, and then test period is used to compare the set of best models (lines 377-380). This should be clarified. In addition, it is not clear if validation is done for the hyperparameters selection (the 5 setups mentioned at line 372) or between different trained version of the model using the same hyperparameters set. Also, is there first a phase to select the hyperparameters (which require to train and test the model), and then a new training phase, or are they both done at once?
**Answer:**
We hope that the additional paragraph and table in section 2.7 (Experimental setup) clarify these points:
The validation period is used to estimate the performance of a specific set of hyperparameters and therefore choose the best fitting set of hyperparameters. The testing period is never part of any training or parameter or model selection procedure and can therefore be understood as a test of the predictive ability of a model on a new data set. The training/validation split in only applied for the neural networks (FNN, RNN-LSTM, RNN-GRU), while all other models use cross-validation (i.e. multiple training/validation splits are applied and the validation performance is estimated more robustly).

**Review:**
Lines 366-367: Do you mean 60% -> training, 20% -> validation? Please clarify.
**Answer:**
We changed the sentence to: "… was done by using a training/validation split with 60% data for training and 20% data for validation."

**Review:**
Lines 368-369: what is the "standard way of training neural networks…", the 50 times training or your approach?
**Answer:**
We will change the sentence to: "Furthermore, the training/validation split is the standard way of training neural networks for real world applications."

**Review:**

Lines 395-397: Please add some citation to the statements made here.

**Answer:**

Thank you for your suggestion. We changed the lines to: "Due to climate change induced warming trends, both air temperatures and water temperatures are steadily increasing (Mohseni, Erickson and Stefan, 1999; Pedersen and Sand-Jensen, 2007; Harvey *et al.*, 2011; Kędra, 2020). This is clearly visible when comparing the change in number of extreme warm days and the increase of mean water temperature in all studied catchments with time."

References:

Harvey, R. *et al.* (2011) 'The influence of air temperature on water temperature and the concentration of dissolved oxygen in Newfoundland Rivers', *Canadian Water Resources Journal*. Taylor & Francis Group , 36(2), pp. 171–192. doi: 10.4296/cwrj3602849.

Kędra, M. (2020) 'Regional Response to Global Warming: Water Temperature Trends in Semi-Natural Mountain River Systems', *Water*. MDPI AG, 12(1), p. 283. doi: 10.3390/w12010283.

Mohseni, O., Erickson, T. R. and Stefan, H. G. (1999) 'Sensitivity of stream temperatures in the United States to air temperatures projected under a global warming scenario', *Water Resources Research*. John Wiley & Sons, Ltd, 35(12), pp. 3723–3733. doi: 10.1029/1999WR900193.

Pedersen, N. L. and Sand-Jensen, K. (2007) 'Temperature in lowland Danish streams: contemporary patterns, empirical models and future scenarios', *Hydrological Processes*. John Wiley & Sons, Ltd, 21(3), pp. 348–358. doi: 10.1002/hyp.6237.

**Review:**

**Answer:**

We are sorry for this unclear phrasing, we changed it to:

"Since test data consists of the last 20% of the overall data, the exact length of these time series is dependent on the catchment but is always a subset of the years 2008-2015."

**Review:**

Lines 438-439: Do you have any explanation regarding the difference of performance observed?

**Answer:**

Thank you for this interesting question. As mentioned above, the relationship between catchment characteristics and ML model performance is highly interesting and will be addressed in a subsequent publication in detail. However, we think that the general performance of statistical/ML approaches for stream water temperature prediction is most certainly related to a few driving catchment characteristics and believe that we will be able to answer this in the above-mentioned, subsequent paper. This will include an analysis of the relation to specific catchment characteristics and corresponding driving hydrological processes.

**Review:**

Lines 450-455: I'm not completely convinced by the significance of this regression. Indeed, simply from boxplots we do see that the difference between catchments is by far the most important predictor.

**Answer:**

In general, the significance of multiple factors (catchment, experiment, model type) in the regression model does not mean that they are equally important. As you state, the catchment

influence is by far larger than the influence of the other regressors However, from these results we can show that the other two regressors also explain some part of the overall variance. Since we know that the catchment influence is by far the largest, we might include this statement here as well to avoid potential misunderstandings. Thus, we propose to add a sentence in line 455: "Overall, the influence of the catchment is higher than the influence of model type and experiment, which is clearly shown with their around one order of magnitude larger coefficients."

**Review:**
Line 487: What is the number of time steps and optimal time step here?
**Answer:**
The number of time steps is a RNN model hyperparameter which defines the number of previous time steps used as model input. With "optimal time steps" we meant the number of timesteps estimated with the Bayesian hyperparameter optimization. To clarify this, we changed the sentence to:

"By removing time information from the inputs, the estimated time steps by Bayesian hyperparameter optimization are 37.78 days longer than when using time information as additional input."

**Review:**
Lines 489-490: Total time should also be provided in order to see how this ~2h decrease is important. Also, how is the p-value obtained here?
**Answer:**
The added overview of model run times (related to one previous major comment) should enable comparison in the revised manuscript. The p-value is obtained by comparing the distribution of training times of the RNN models with and without time information (fuzzy months) with a Kruskal-Wallis test.

**Review:**
Line 536: The claim about short term predictions and CC is too ambitious here. What is shown is the increase in performance compared to benchmark models, which is already a really important step. For short term prediction and CC, see my longer comments above, but I think more work and discussion are needed to really assess the ability of the models for these applications.
**Answer:**
We agree. Please find the related changes in the revised manuscript in the comments 1: 3 and 9:11 above. We also kindly refer to our next paper at this point.

**Review:**
Lines 546-547: The improvement here can be explained by the Bayesian optimization method used?
**Answer:**
While an adequate hyperparameter optimization was necessary for some models (e.g. FNN, XGBoost), others would also produce good results without it (e.g. RF). Thus, the improvement can be attributed to the combination of the adequate representation of time (fuzzy months) as data input, the applied hyperparameter optimization, the choice of lagged time steps and the used input variables. To clarify this, we changed the statement to:
"Consequently, our presented approaches show a significant improvement compared to existing machine learning daily stream water temperature prediction models, which can be attributed to the adequate representation of time (fuzzy months) as data input, the applied hyperparameter optimization, the choice of lagged time steps and the used input variables."

**Review:**

Lines 548-550: what do you mean by "spatial information at different scale"? Indeed, ML models do not provide any spatially distributed output (which can be achieved with distributed physical models), but only point informations.

**Answer:**

Here we refer to the ability of ML models to generally being able to learn from given sequences (e.g. time series) or objects (e.g. ortho pictures). By writing "spatial information" we mean that catchment characteristics from objects could potentially be used too. We revised the statement to make it clearer and changed it to: "However, machine learning methods are more powerful and flexible than previous modelling approaches and are able to simultaneously use spatial and temporal information at different scales (Reichstein et al., 2019)."

---

## Referee Report (RR1)

**Pr. Salim Heddam**
*heddamsalim@yahoo.fr.*
https://orcid.org/0000-0002-8055-8463

The authors have correctly and seriously addressed the necessary amendments and the manuscript was significantly improved and scientifically sounds. There are no needs for any other improvement and the actual form is acceptable.